# Atomic-layer Rashba-type superconductor protected by dynamic spin-momentum locking

Shunsuke Yoshizawa [1]✉, Takahiro Kobayashi[2], Yoshitaka Nakata[3], Koichiro Yaji[4,11], Kenta Yokota [5,6], Fumio Komori[4], Shik Shin [4,7], Kazuyuki Sakamoto [3,8,9,10] & Takashi Uchihashi [5,6]✉

Spin-momentum locking is essential to the spin-split Fermi surfaces of inversion-symmetry broken materials, which are caused by either Rashba-type or Zeeman-type spin-orbit coupling (SOC). While the effect of Zeeman-type SOC on superconductivity has experimentally been shown recently, that of Rashba-type SOC remains elusive. Here we report on convincing evidence for the critical role of the spin-momentum locking on crystalline atomic-layer superconductors on surfaces, for which the presence of the Rashba-type SOC is demonstrated. In-situ electron transport measurements reveal that in-plane upper critical magnetic field is anomalously enhanced, reaching approximately three times the Pauli limit at $T = 0$. Our quantitative analysis clarifies that dynamic spin-momentum locking, a mechanism where spin is forced to flip at every elastic electron scattering, suppresses the Cooper pair-breaking parameter by orders of magnitude and thereby protects superconductivity. The present result provides a new insight into how superconductivity can survive the detrimental effects of strong magnetic fields and exchange interactions.

[1] Research Center for Advanced Measurement and Characterization, National Institute for Materials Science, Tsukuba, Ibaraki, Japan. [2] Department of Material and Life Science, Osaka University, Suita, Osaka, Japan. [3] Department of Materials Science, Chiba University, Inage-ku, Chiba, Japan. [4] Institute for Solid State Physics, The University of Tokyo, Kashiwa, Chiba, Japan. [5] International Center for Materials Nanoarchitectonics (WPI-MANA), National Institute for Materials Science, Tsukuba, Ibaraki, Japan. [6] Department of Condensed Matter Physics, Graduate School of Science, Hokkaido University, Sapporo, Hokkaido, Japan. [7] Office of University Professor, The University of Tokyo, Kashiwa, Chiba, Japan. [8] Center for Spintronics Research Network, Graduate School of Engineering Science, Osaka University, Toyonaka, Osaka, Japan. [9] Department of Applied Physics, Osaka University, Suita, Osaka, Japan. [10] Molecular Chirality Research Center, Chiba University, Inage-ku, Chiba, Japan. [11]Present address: Research Center for Advanced Measurement and Characterization, National Institute for Materials Science, Tsukuba, Ibaraki, Japan. ✉email: YOSHIZAWA.Shunsuke@nims.go.jp; UCHIHASHI.Takashi@nims.go.jp

The breaking of the out-of-plane or in-plane inversion symmetry in two-dimensional (2D) systems gives rise to Rashba-type or Zeeman-type spin–orbit coupling (SOC), respectively, which plays important roles in spintronics, valley-tronics, optoelectronics and superconductivity[1–5]. Both types of SOCs cause the Fermi surface to be spin-split and the spin-momentum relation to be locked, but the spin polarisation in the momentum space is distinctively different; Rashba-type SOC forces the spins to be polarised in the in-plane direction while Zeeman-type SOC in the out-of-plane direction (Fig. 1a, b)[1,2]. These unique spin structures have notable implications in terms of superconductivity under strong magnetic fields[6–13].

Suppose the magnetic field is applied to a 2D superconductor precisely in the in-plane direction. Since electron orbitals are barely affected in this configuration, Cooper pairs are destroyed mainly due to the field-induced parallel alignment of the electron spins, which otherwise form an anti-parallel spin-singlet state. This mechanism is called paramagnetic pair breaking, and the upper critical magnetic field $B_{c2\parallel}$ determined by this effect is called the Pauli limit $B_{Pauli}$[14,15]. In the presence of Zeeman-type SOC, the spins are hardly tilted in the field direction because they are statically locked in the out-of-plane direction. This suppresses the paramagnetic pair breaking effect and substantially enhances $B_{c2\parallel}$ over $B_{Pauli}$[8,9]. By contrast, the in-plane spin-momentum locking due to Rashba-type SOC can enhance $B_{c2\parallel}$ only by a factor of $\sqrt{2}$ because of a significant deformation of the Fermi surfaces due to the locking[6]. Nevertheless, Rashba-type SOC may also strongly enhance $B_{c2\parallel}$ if a dynamic electron scattering process is involved. In this case, because of the spin-momentum locking, the spin is forced to flip at every momentum change accompanied by elastic scattering (Fig. 1c). The mechanism, referred to as dynamic spin-momentum locking here, should cause frequent spin scatterings while preserving the time-reversal symmetry. This enhances $B_{c2\parallel}$ through the suppression of paramagnetic pair-breaking effect even in crystalline systems in an analogous manner as the conventional spin–orbit scattering does in disordered systems. Although such an effect was suggested by Nam et al. for Pb thin films, it was considered dominated by the orbital pair-breaking and hence has remained elusive[10]. Furthermore, the presence of the Rashba-type SOC itself was an assumption, and the material may include Zeeman-type SOC[11]. Experimentally resolving this problem requires one to confirm the exclusive presence of the Rashba-type SOC in a relevant system. It is also important to investigate its superconducting properties under a controlled environment to avoid any extrinsic effects.

In the present study, we adopt a crystalline In atomic-layer on a Si(111) surface [Si(111)-($\sqrt{7}\times\sqrt{3}$)-In] to fulfil this requirement. Clear Fermi surface splitting and in-plane spin polarisation are demonstrated by angle-resolved photoemission spectroscopy (ARPES) and density functional theory (DFT) calculations, confirming the exclusive presence of the Rashba-type SOC. In situ electron transport measurements under ultrahigh vacuum (UHV) environment reveal that $B_{c2\parallel}$ is anomalously enhanced over the Pauli limit $B_{Pauli}$. The enhancement factor defined by $B_{c2\parallel}/B_{Pauli}$ reaches ~3 and exceeds the factor of $\sqrt{2}$ expected for the static locking effect of the Rashba-type SOC. Our quantitative data analysis clarifies that the paramagnetic pair-breaking parameter $\alpha_P$ is strongly suppressed by orders of magnitudes from the value estimated for the conventional spin–orbit scattering. The spin scattering times $\tau_s$ directly related to $\alpha_P$ are in satisfactory agreement with the electron elastic scattering times $\tau_{el}$, proving the idea of spin flipping at every momentum change. These results provide compelling evidence that this 2D superconductor with Rashba-type SOC is protected by dynamic spin-momentum locking.

## Results

**Rashba-type SOC revealed by ARPES and DFT.** The Si(111)-($\sqrt{7}\times\sqrt{3}$)-In (referred to as $\sqrt{7}\times\sqrt{3}$-In here) consists of a uniform In bilayer covering the Si(111)-1×1 surface with a periodicity of $\sqrt{7}\times\sqrt{3}$ (Fig. 2a)[16,17], and superconductivity occurs below 3 K[18,19]. The breaking of the out-of-plane inversion symmetry due to the presence of the Si surface leads to the Rashba-type SOC as described below, as reported for other atomic-layer crystals on surfaces[20–23]. The material has highly dispersed electronic bands and simple chemical composition without magnetic or heavy elements[24], which allows us to neglect complex correlation effects. Despite these ideal features, studying superconducting properties of $\sqrt{7}\times\sqrt{3}$-In is challenging because

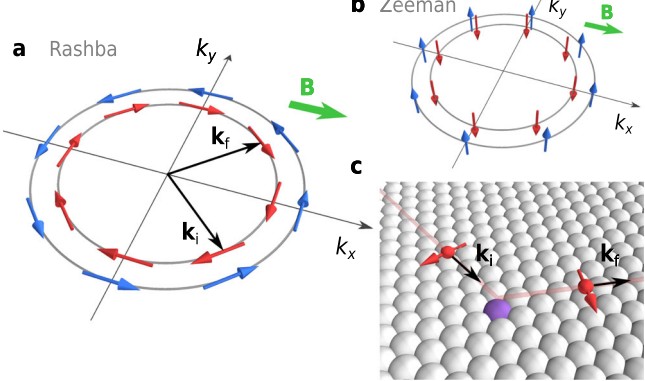

**Fig. 1 Schematic illustration of dynamic spin-momentum locking. a** Fermi surfaces in the presence of Rashba-type SOC. The spins are polarised in the in-plane directions and locked to the momentum. The split Fermi surfaces are characterised by spin textures with opposite helicities. **b** Fermi surfaces in the presence of Zeeman-type SOC. The spins are oriented in the out-of-plane directions. The green arrows in (**a**) and (**b**) indicate a magnetic field applied in the in-plane direction. **c** When an electron at the initial state $\mathbf{k}_i$ is elastically scattered to $\mathbf{k}_f$ by a non-magnetic scattering centre (depicted by a purple ball), its spin is forced to rotate.

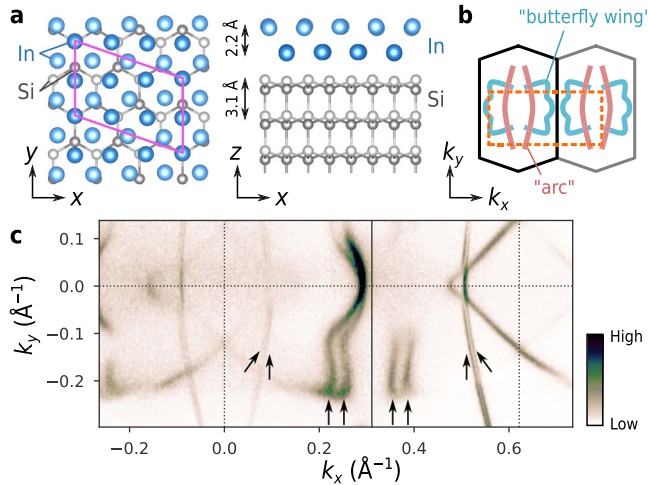

**Fig. 2 Crystal structure and experimentally-observed Fermi surface. a** Top view (left) and side view (right) of the crystal structure of $\sqrt{7}\times\sqrt{3}$-In. The x, y, and z axes are defined to be [1$\bar{1}$0], [11$\bar{2}$] and [111] directions, respectively. **b** Schematic of the 1st and 2nd Brillouin zones. The dashed box represents the area of the photoelectron intensity map in (**c**). The characteristic arc feature and butterfly-wing feature of the Fermi surface are depicted. **c** Photoelectron intensity map at $E_F$. The arrows indicate the portions of the Fermi surface with a clear splitting.

the susceptibility to foreign molecules and surface defects prohibits air exposure and the usage of conventional cryogenic and high-magnetic-field systems[25,26]. In this study, all experiments, including the transport measurements, were performed in UHV to eliminate the possibility of sample degradation (see "Materials and Methods").

The details of the electronic structures and the presence of Rashba-type SOC are clarified through ARPES measurements and DFT calculations. Figure 2c shows the photoelectron intensity map at the Fermi energy ($E_F$) measured over the momentum-space region depicted in Fig. 2b. While the result is consistent with the previous studies[16,24], it clearly resolves the splitting of the Fermi surfaces for the first time, which is particularly conspicuous on the "arc" and "butterfly-wing" portions (see the pairs of arrows). This finding was fully reproduced by our DFT calculations. The computed Fermi surface structure is essentially identical to the ARPES data (Fig. 3a). While the magnitude of the energy splitting at $E_F$ ($\Delta_R$) is small along the high-symmetry lines (Y–Γ–X), it is larger at the butterfly-wing along the P–Q line (Fig. 3f). Figure 3c shows that the distribution of $\Delta_R$ exhibits a peak around 15–20 meV and ranges up to 90 meV. The DFT calculations also confirm that these Fermi surfaces are indeed spin-polarised. As indicated by the arrows in Fig. 3a, the spins are oriented in the in-plane directions, as expected from Rashba-type SOC. The effect of Zeeman-type SOC is negligible, judging from the fact that the out-of-plane components of spins are nearly absent (Fig. 3d). This point will be discussed later in detail. Interestingly, the azimuthal orientation of the spins on the butterfly wing features deviates from the helical spin texture characteristic of the standard Rashba-type SOC. This in-plane spin texture does not affect the conclusion of the present study, and its microscopic origin will be discussed elsewhere[27]. We also mention the presence of a large anisotropy in the Fermi velocity $\mathbf{v}_F$, which was computed as the gradient of band dispersion (Fig. 3b). The histogram in Fig. 3e shows that $|\mathbf{v}_F|$ ranges from $2 \times 10^5$ to $1.5 \times 10^6$ m s$^{-1}$. The detailed band structure information obtained here will be used later.

**Robust superconductivity in in-plane magnetic fields**. Six $\sqrt{7} \times \sqrt{3}$-In samples were prepared for electron transport experiments. In addition to three nominally flat Si(111) surfaces (Flat#1/#2/#3), we used three vicinal surfaces (Vicinal#1/#2 with a miscut angle of 0.5° and Vicinal#3 with a miscut angle of 1.1°) to control the density of scattering sources. These sample surfaces consisted of atomically flat terraces separated by steps, as observed by scanning tunnelling microscopy (STM) (Fig. 4a: Flat#1, Fig. 4b: Vicinal#1). The low-energy electron diffraction (LEED) patterns of Flat#1 and Vicinal#1 confirmed the exclusive presence of $\sqrt{7} \times \sqrt{3}$ structures with multi- and single-domains, respectively (Insets of Fig. 4a, b). Figure 4c shows the temperature ($T$) dependence of sheet resistance ($R_{sheet}$) recorded at zero magnetic field. The curves of the other four samples are available in Supplementary Fig. 1. All of the samples exhibit sharp superconducting transitions at $T_{c0}$, while precursors due to the 2D fluctuation effects are evident at $T > T_{c0}$[28]. Here $T_{c0}$ is defined as the Bardeen–Cooper–Schrieffer (BCS) mean-field critical temperature, which was determined by fitting to an empirical formula[29] (see Supplementary Note 1). The same fitting procedure also gives normal sheet resistance $R_n$ in the absence of the 2D fluctuation effects. The obtained parameters for $T_{c0}$ and $R_n$ are presented in Table 1. The small $R_n$ of 36–90 Ω reflects the high crystallinity of the samples. These values are comparable to those reported for transition-metal dichalcogenide samples used in the studies of Zeeman-type SOC[8,9].

We now focus on the effects of strong magnetic fields on superconductivity of $\sqrt{7} \times \sqrt{3}$-In. Figure 5a shows the

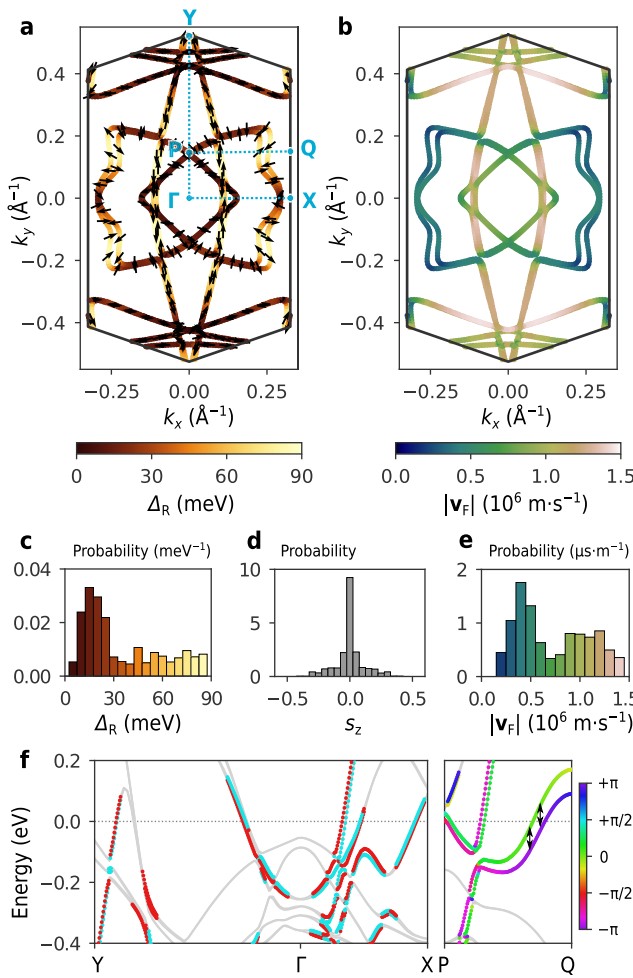

**Fig. 3 Computed spin-split Fermi surface and Fermi velocity. a** Fermi surface obtained from the DFT calculation. The colour indicates the magnitude of band splitting $\Delta_R$. Here, $\Delta_R$ is defined at each point on the Fermi surface as the energy difference from the partner band, as depicted by the double-headed arrows in (**f**). The small arrows indicate the orientation of the spins. **b** Fermi velocity $|\mathbf{v}_F|$ computed from the band dispersion. **c–e** Histograms of $\Delta_R$ (**c**), the z component of the spin, $s_z$, (**d**), and $|\mathbf{v}_F|$ (**e**) measured on the Fermi surface. These histograms reflect the weighting factor to the density of states given by $dl/|\mathbf{v}_F|$, where $dl$ is the line segment on the Fermi surface. **c, e** share the same colour scales as (**a**) and (**b**), respectively. **f** Energy bands along Y–Γ–X and P–Q. The vertical axis shows the energy measured from the Fermi level. The colour indicates the relative direction of spin with respect to momentum; red and cyan correspond to the clockwise and counterclockwise helicities. For clarity, the states with $\Delta_R < 10$ meV are coloured in grey.

temperature dependence of sheet resistance $R_{sheet}$ of Vicinal#1 measured under magnetic fields, which were applied precisely in the in-plane direction. The data of the other samples are presented in Supplementary Figure 2. While slight shifts and broadenings of the resistive transition were detected, superconductivity persisted even at the maximum magnetic field of $B = 5$ T. Figure 5c shows the magnetic field dependence of $T_c$ of all six samples, where $T_c$ is determined from $T$ at which $R_{sheet}$ decreases to half of $R_n$. The data show that the lowering of $T_c$ as a function of $B$ is quadratic and reaches 23% of $T_{c0}$ at 8.25 T for Flat#3. By contrast, for out-of-plane magnetic fields, the superconducting transition was rapidly suppressed and disappeared above $B = 0.5$ T (Fig. 5b). The lowering of $T_c$ as a function of $B$ is linear (Fig. 5d). Our detailed analysis for out-of-plane upper

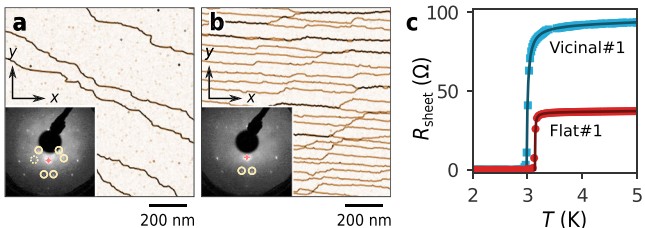

**Fig. 4 Characterisation of samples for electron transport measurements.** **a**, **b** STM images of Flat#1 and Vicinal#1, respectively. The derivative (gradient) of the topographic data is displayed for highlighting the locations of atomic steps. The insets are the corresponding LEED patterns taken at a 94 eV beam energy. The circles indicate the peaks reflecting the $\sqrt{7} \times \sqrt{3}$ periodicity. The plus symbol indicates (0, 0). **c** Resistance curves of Flat#1 and Vicinal#1 measured at zero magnetic field. The solid curves are fitting by an empirical formula (See Supplementary Note 1).

critical field $B_{c2\perp}$ shows that the observed rapid quenching of superconductivity is due to penetration of vortices, i.e. to orbital pair-breaking effect (see Supplementary Fig. 3 and Supplementary Note 2). The robust superconductivity against the in-plane fields, in contrast, indicates that the pair-breaking is not caused by the orbital effect but rather by the paramagnetic effect as expected. For the present superconductor with $T_{c0} = 2.97$–$3.14$ K, the Pauli limit $B_{Pauli}$ is equal to 5.5–5.8 T from the relation $B_{Pauli} = 1.86$ (T K$^{-1}$)$T_{c0}$[14,15]. Since the observed $B_{c2\parallel}$ apparently exceeds this limit as $T \to 0$, the paramagnetic pair breaking effect must be substantially suppressed.

**Paramagnetically limited upper critical field**. It is widely known that spin scattering is induced occasionally at an elastic electron scattering event by the atomistic SOC. In the presence of this conventional spin–orbit scattering, Cooper pairs are no longer exact spin-singlet states. It induces a finite spin susceptibility in the system and lowers the Zeeman energy gain acquired by breaking a Cooper pair under a magnetic field, thus suppressing the paramagnetic pair-breaking effects[30]. Here we assume this mechanism and, without taking account of the Rashba-type SOC, analyse the magnetic field effects on superconductivity in terms of pair-breaking parameters. The dependence of $T_c$ on magnetic field **B** can be described using a universal function given by

$$\ln\left(\frac{T_c}{T_{c0}}\right) = \psi\left(\frac{1}{2}\right) - \psi\left(\frac{1}{2} + \frac{\alpha(\mathbf{B})}{2\pi k_B T_c}\right), \quad (1)$$

where $\psi$ is the digamma function, and $\alpha(\mathbf{B})$ denotes field-dependent pair-breaking parameter[31]. $\alpha(\mathbf{B})$ is the sum of three contributions: $\alpha_{O\perp}$ and $\alpha_{O\parallel}$ representing the orbital effects due to out-of-plane ($B_\perp$) and in-plane ($B_\parallel$) fields and $\alpha_P$ the paramagnetic effect due to the total field $|\mathbf{B}|$ in the presence of frequent spin scatterings. It is given by the equation

$$\alpha(\mathbf{B}) = \alpha_{O\perp} + \alpha_{O\parallel} + \alpha_P \quad (2)$$

$$= c_{O\perp} B_\perp + c_{O\parallel} B_\parallel^2 + c_P |\mathbf{B}|^2, \quad (3)$$

where $c_{O\perp}$, $c_{O\parallel}$ and $c_P$ are coefficients for individual contributions[32]. This form of the pair-breaking parameter is closely related to the Klemm-Luther-Beasley (KLB) model proposed for 2D superconductors with conventional spin–orbit scattering[33,34].

In the present study, the addition of the $\alpha_{O\parallel}$ term allows us to account for the orbital effect within the superconducting layer under the in-plane magnetic field, which is not included in the KLB model. This effect played a crucial role in few-layer Pb films studied previously[10]. For the in-plane configuration, $B_\perp \simeq \theta_e|\mathbf{B}|$ and $B_\parallel \simeq |\mathbf{B}|$, where $\theta_e$ is the angular error. All coefficients were determined by fitting Eq. (1) to the experimental data in Fig. 5c,

**Table 1 List of parameters obtained for the six samples.**

| Sample (fs) | miscut (deg.) | $R_n$ (Ω) | $T_{c0}$ (K) | $B_{Pauli}$ (T) | $\xi$ (nm) | $c_{O\perp}$ (meV·T⁻¹) | $c_{O\parallel}$ | $c_P$ (μeV·T⁻²) | $\theta_e$ (μeV·T⁻²) | $\tau_s$ (deg.) | $\tau_{el}$ (fs) |
|---|---|---|---|---|---|---|---|---|---|---|---|
| Flat#1 | 0 | 35.9 | 3.14 | 5.83 | 41.3 (2.9) | 1.67 (0.09) | 0.086 (0.005) | 0.66 (0.09) | 0.07 (0.02) | 86 (12) | 71.7 |
| Flat#2 | 0 | 48.3 | 3.14 | 5.83 | 41.4 (2.7) | 1.68 (0.08) | 0.086 (0.004) | 0.39 (0.11) | 0.12 (0.02) | 52 (14) | 53.3 |
| Flat#3 | 0 | 53.2 | 3.11 | 5.78 | 42.9 (1.9) | 1.85 (0.16) | 0.095 (0.008) | 0.25 (0.14) | 0.21 (0.03) | 33 (18) | 44.9 |
| Vicinal#1 | 0.5 | 90.0 | 2.99 | 5.55 | 29.2 (0.9) | 0.83 (0.02) | 0.042 (0.001) | 0.53 (0.09) | 0.08 (0.03) | 69 (12) | 28.6 |
| Vicinal#2 | 0.5 | 65.7 | 3.04 | 5.64 | 30.3 (0.9) | 0.90 (0.02) | 0.046 (0.001) | 0.53 (0.09) | 0.04 (0.03) | 70 (12) | 39.2 |
| Vicinal#3 | 1.1 | 82.1 | 2.97 | 5.52 | 31.9 (0.6) | 0.99 (0.04) | 0.051 (0.002) | 0.44 (0.09) | 0.06 (0.03) | 57 (12) | 31.4 |

$\xi$ is the Ginzburg-Landau coherence length estimated from the data in out-of-plane magnetic fields (See Supplementary Note 2). The values in the parentheses are the estimates of errors propagated from the accuracy of the calibration curve for magnetoresistance of the temperature sensor (0.005 K for $B \le 5$ T and 0.04 K for $B \ge 5$ T) and the hysteresis of the superconducting magnet (0.004 T).

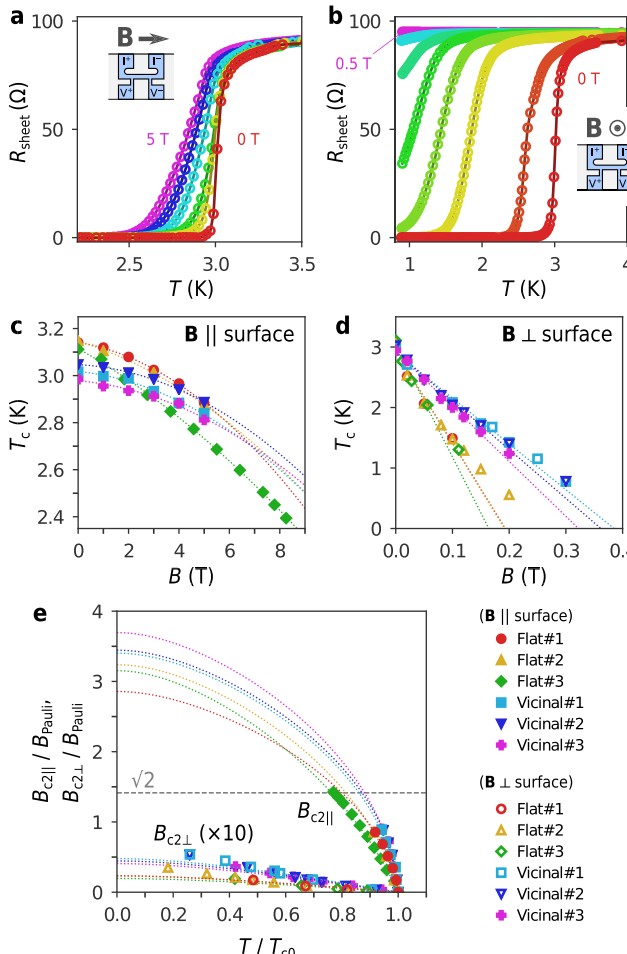

**Fig. 5 Superconducting properties in in-plane and out-of-plane magnetic fields. a** Resistance curves of Vicinal#1 in in-plane magnetic fields of $B = 0$, 1, 2, 3, 4 and 5 T. **b** Resistance curves of Vicinal#1 in out-of-plane magnetic fields of $B = 0$, 0.02, 0.1, 0.15, 0.2, 0.25, 0.3 and 0.5 T. The insets of (**a**) and (**b**) show the directions of the magnetic fields with respect to the samples. The samples were patterned in a shape suitable for four-terminal measurements, as represented by areas coloured in light blue. $I^{+(-)}$ and $V^{+(-)}$ are the current and voltage terminals, respectively. **c** Field dependence of $T_c$ in in-plane magnetic fields. **d** Field dependence of $T_c$ in out-of-plane magnetic fields. The dotted curves in (**c**) and (**d**) are the quadratic and linear functions fitted to the data. **e** Comparison of $B_{c2\parallel}$ and $B_{c2\perp}$. For clarity, $B_{c2\perp}$ is scaled by a factor of 10. The dotted curves are the universal function Eq. (1) plotted with parameters determined from the fitting analyses. The dashed horizontal line indicates the enhancement factor $\sqrt{2}$ for static locking effect of Rashba-type SOC. The relatively large variation in the fitting curves for $B_{c2\parallel}$ originates mainly from the angular error of the sample orientation denoted by $\theta_e$ in the main text.

d, and the results are listed in Table 1 (for details, see "Materials and Methods"). From the values of $c_{O\perp}$, $c_{O\parallel}$, $c_P$, we conclude $\alpha_{O\perp}$, $\alpha_{O\parallel} \ll \alpha_P$ in the in-plane configuration, meaning that the pair breaking is dominated by the paramagnetic effect. This is distinct from the finding by Nam et al. that the orbital effect is the primary pair breaking mechanism for 5–13 Pb monolayers on the Si(111) surface[10]. Figure 5e plots $B_{c2\parallel}/B_{Pauli}$ and $B_{c2\perp}/B_{Pauli}$ as a function of $T_c/T_{c0}$, along with their extrapolations down to $T = 0$ calculated with the universal function of Eq. (1). $B_{c2\parallel}/B_{Pauli}$ is found to reach ~ 3 at $T = 0$. We note that this enhancement factor exceeds the value of $\sqrt{2}$, which is expected for the static effect of Rashba-type spin momentum locking. This claim is directly

evidenced by the maximum value of $B_{c2\parallel}/B_{Pauli} = 1.43$ obtained for Flat#3.

**Spin flipping rate enhanced by dynamic spin-momentum locking.** The strong enhancement of $B_{c2\parallel}$ observed above is actually not attributed to the atomistic SOC, but to the Rashba-type SOC as explained in the following. We first estimate elastic scattering time $\tau_{el}$ from the normal-state sheet resistance $R_n$. The calculation was carried out by explicitly considering the anisotropy of Fermi velocity $\mathbf{v}_F$ computed above (Fig. 3c, d) and by employing the Boltzmann theory under relaxation approximation[35]. The sheet conductance is given by

$$\sigma_{\mu\mu} = \tau_{el}I_{\mu\mu} \qquad (4)$$

with

$$I_{\mu\mu} \equiv \frac{e^2}{(2\pi)^2\hbar}\int_{FS} dk \frac{v_{F\mu}(\mathbf{k})^2}{|\mathbf{v}_F(\mathbf{k})|}, \qquad (5)$$

where $v_{F\mu}$ ($\mu = x$ or $y$) is the $\mu$ component of $\mathbf{v}_F$. The integral was taken over all the spin-split Fermi surfaces, yielding $I_{xx} = 3.9 \times 10^{-4}\ \Omega^{-1}\ \text{fs}^{-1}$ and $I_{yy} = 4.5 \times 10^{-4}\ \Omega^{-1}\ \text{fs}^{-1}$. $\tau_{el}$ was evaluated from $R_n^{-1} = (\sigma_{xx} + \sigma_{yy})/2$ for multi-domain flat samples (Flat#1/#2/#3) and from $R_n^{-1} = \sigma_{xx}$ for single-domain vicinal samples (Vicinal#1/#2#3). This gives $\tau_{el} = 71.7$, 53.3, 44.9 fs for Flat#1/#2/#3 and $\tau_{el} = 28.6$, 39.2, 31.4 fs for Vicinal#1/#2/#3, respectively (Table 1). We then estimate the spin scattering time $\tau_s$ from the coefficient $c_P$ for paramagnetic pair breaking effect. $\tau_s$ is calculated with an equation

$$c_P = \frac{3\tau_s\mu_B^2}{2\hbar}, \qquad (6)$$

where $\mu_B$ is the Bohr magneton and $\hbar$ the reduced Plank constant[36]. This gives $\tau_s = 86 \pm 12$, $52 \pm 14$, $33 \pm 18$ fs for Flat#1/#2/#3, and $\tau_s = 69 \pm 12$, $70 \pm 12$, $57 \pm 12$ fs for Vicinal#1/#2/#3 (see Table 1). These results lead to $\tau_{el}/\tau_s \simeq 0.5 - 1$. Nevertheless, if only the conventional spin–orbit scattering is considered, $\tau_s$ should be much larger than the $\tau_{el}$. In this case, the ratio $\tau_{el}/\tau_s$ should be on the order of $(Z\alpha)^4$, where $Z$ is the atomic number and $\alpha$ is the fine structure constant[30]. For In ($Z = 49$), $\tau_{el}/\tau_s \sim 1/60$. An experimental study reported an even smaller $\tau_{el}/\tau_s$ of about $10^{-3}$ for thin In films[37]. Therefore, the spin–orbit scattering that occurs in the absence of the Rashba-type SOC cannot account for our result. In contrast, if the Rashba-type SOC is considered, it can be reasonably explained based on the concept of dynamic spin-momentum locking; namely, every elastic scattering should contribute a spin flipping and $\tau_{el}/\tau_s$ approaches unity. The decrease in $\tau_s$ together with Eq. (3) and Eq. (6) means the paramagnetic pair breaking parameter $\alpha_P$ is suppressed by orders of magnitude from the value expected for the conventional spin–orbit scattering.

Remarkably, for the flat samples, $\tau_s$ falls equal to $\tau_{el}$ within the experimental error. By contrast, $\tau_s$ is larger than $\tau_{el}$ by a factor of two for the vicinal samples. This can be reasonably explained by an energy broadening caused by electron elastic scattering, $\hbar/\tau_{el}$. For vicinal samples, $\hbar/\tau_{el} = 16$–24 meV is comparable to the peak energy in the distribution of $\Delta_R$ (see Fig. 3b). This energy broadening degrades the spin polarisation at a large portion of the Fermi surface and partially unlocks the spin-momentum relation, resulting in a recovery of spin scattering time $\tau_s$. For flat samples, $\hbar/\tau_{el} = 9$–14 meV $< \Delta_R$, meaning that the spin texture of the energy bands remains intact for the whole Fermi surface. This argument further supports our conclusion on the critical role of the dynamic effect of the Rashba-type SOC.

Finally, we note that the static spin-momentum locking due to the Rashba-type SOC can enhance the in-plane critical field $B_{c2\parallel}$ by

a factor of $\sqrt{2}$ from the Pauli limit. This effect is likely to be weakened by electron scattering and mixing between different spin states, but here we estimate the upper limit of error in spin scattering time $\tau_s$ (for a detailed discussion, see Supplementary Note 3). When it is taken into account as an effective magnetic field $B_{eff} = (1/\sqrt{2})B$, the value of $\tau_s$ obtained above is doubled, leading to $\tau_{el}/\tau_s = 0.25-0.5$. These values are still much higher than 1/60-1/1000 expected from the atomistic spin–orbit scattering mechanism. Therefore, the result is not attributable only to the conventional mechanism, and our conclusion remains the same.

## Discussion

Here we discuss the consistency with the theoretical studies of Rashba-type superconductors with non-magnetic impurities[10,38–40]. These studies predict that upper critical field increases with the decrease in elastic scattering time $\tau_{el}$. In 2D, the enhancement factor corresponds to a pair-breaking parameter $\alpha = (2\mu_B^2 \tau_{el}/\hbar)B^2$ in the limit of strong SOC ($\hbar/\tau_{el} \ll \Delta_R$)[10]. This expression is equivalent to Eq. (6) if $\tau_s$ is replaced by $(4/3)\tau_{el}$. The agreement allows us to interpret the above theoretical result in terms of dynamic spin-momentum locking. Theories also claim that the ground state of a 2D superconductor with Rashba-type SOC has a helical state with a spatially modulated order parameter[38–40]. The formation of the helical state may increase $B_{c2\parallel}$, and a previous study on a quench-condensed monolayer Pb film attributed their observation of giant $B_{c2\parallel}$ to this effect[7]. However, the enhancement factor is only in the order of $\sim (\Delta_R/E_F)^2$ and is usually negligible because $\Delta_R \ll E_F$[40]. Therefore, the observed large $B_{c2\parallel}$ in the present and previous studies are not attributable to the formation of the helical state.

Another issue to be discussed is the possible effect of a finite Zeeman-type SOC, which is suggested from the non-zero out-of-plane spin polarisations shown in Fig. 3d. From the spin polarisation direction calculated as a function of energy splitting, one sees that the spins align in the in-plane directions for the most of energy regions (Supplementary Figure 6). The spins tend to tilt toward the out-of-plane direction below 30 meV, but the off-angle is about 45° at most. Namely, there is no region where the Zeeman-type SOC is dominant. This non-dominant Zeeman SOC confined to the small area of the Fermi surface can barely enhance $B_{c2\parallel}$ because the enhancement factor is determined by an average over the whole Fermi surface[41]. If the dynamics of spins is considered, the effect of the Zeeman-type SOC can be suppressed even more. Thus, we conclude that the Zeeman-type SOC plays only a minor role in the present system. For more discussions, see Supplementary Note 4.

The present result has significant implications in terms of robustness of a superconductor with the Rashba-type SOC in general under a strong magnetic field as well as in the proximity of a ferromagnet. The presence of a strong exchange interaction at the interface with a ferromagnet usually leads to the destruction of superconductivity to the depth of the coherence length. However, since the destruction of superconductivity by exchange interaction is caused by the same paramagnetic pair-breaking effect[32], the dynamic spin-momentum locking revealed here may help superconductivity to persist even in this situation. This makes realistic the coexistence of a 2D superconductor and a ferromagnet at atomic scales, which has been proposed to realise emergent phenomena such as chiral topological super-conductivity[42–44]. The new insight into the spin-momentum locking obtained in the present study will form the basis of such an unexplored realm of research.

## Methods

**ARPES.** The high-resolution ARPES experiment was conducted in a UHV environment with a base pressure better than $1 \times 10^{-8}$ Pa. The substrate was cut from an n-type (resistivity $\rho < 0.001\ \Omega$ cm) vicinal Si(111) wafer with a miscut angle of 0.5° in the [$\bar{1}\bar{1}2$] direction. The $\sqrt{7} \times \sqrt{3}$-In surface was prepared by thermal deposition of In onto a clean Si(111)-7 × 7 surface followed by annealing at 600 K for 2 min. The sample quality was confirmed from the sharp spots with low background intensity in the LEED patterns. The photoelectrons were excited by a vacuum-ultraviolet laser ($h\nu = 6.994$ eV) and were collected by a hemispherical photoelectron analyser[45]. The sample temperature was maintained at 35 K during the ARPES measurement. The energy and momentum resolutions were 3 meV and $1.4 \times 10^{-3}$ Å$^{-1}$, respectively.

**DFT.** The DFT calculations were performed using the Quantum ESPRESSO package[46]. We employed the augmented plane wave method and used the local density approximation (LDA) for the exchange correlation. The crystal structure of $\sqrt{7} \times \sqrt{3}$-In was modelled by a repeated slab consisting of an In bilayer, six Si bilayers, a H layer for termination, and a vacuum region of thickness 3 nm. We used a cutoff energy of 680 eV for the wave functions and a $6 \times 8 \times 1$ k-point mesh for the Brillouin zone. The geometry optimisation was performed without the SOC until all components of all forces became less than $2.6 \times 10^{-3}$ eV·Å$^{-1}$. Based on the optimised structure, we performed band calculations that included the SOC. To check the reproducibility of our result, we carried out the same calculation from scratch using another DFT package OpenMX[47,48]. The result by OpenMX is essentially the same as the one by Quantum ESPRESSO (See Supplementary Figs. 5, 6, as well as Supplementary Note 5).

**Electron transport.** For transport experiments, six samples were grown on substrates cut from Si(111) wafers (3 mm × 8 mm × 0.38 mm) with miscut angles of 0° (Flat#1, Flat#2, and Flat#3), 0.5° (Vicinal#1 and Vicinal#2), and 1.1° (Vicinal#3) in the [$\bar{1}\bar{1}2$] direction. The non-doped wafers ($\rho \gtrsim 1000\ \Omega$ cm) were chosen so that the electron conduction in bulk can be ignored at low temperatures. The $\sqrt{7} \times \sqrt{3}$-In surface was prepared under the UHV condition (base pressure $1 \times 10^{-8}$ Pa) by depositing In onto a clean Si(111)-7 × 7 surface followed by annealing at 600 K for 10 s. The samples were then characterised by LEED and STM. The current path was defined by Ar$^+$ sputtering using a shadow mask technique[19,29]. Electric contact was made at room temperature by mildly pressing four Au-plated spring probes. The samples were then cooled down to ~ 0.9 K or to ~ 0.4 K by pumping condensed $^4$He or $^3$He with a charcoal sorption pump. The magnetic fields were applied with a superconducting solenoid magnet. The maximum field was 5 T in the experiment of Flat#1/#2 and Vicinal#1/#2/#3 and was 8.25 T in the experiment of Flat#3. The sample was rotated in-situ to tune the angle of the magnetic fields with respect to the sample. The parallel field alignment was judged within an accuracy better than 0.1° from the minimum of sample resistance measured at a constant temperature near the $T_c$. The sample temperature was measured with a Cernox thermometer calibrated in magnetic fields. The DC resistance of the samples was measured using a nanovoltmeter (Keithley 2182A) with a bias current of 1 $\mu$A generated by a source meter (Keithley 2401).

**Fitting analysis of pair-breaking effects.** In the following, we denote $|\mathbf{B}|$ as $B$ for simplicity. The pair-breaking parameters for orbital effects are given by

$$\alpha_{O\parallel} = \frac{De^2\delta^2 B_\parallel^2}{6\hbar} \equiv c_{O\parallel} B_\parallel^2 \qquad (7)$$

for the in-plane component, and

$$\alpha_{O\perp} = DeB_\perp \equiv c_{O\perp} B_\perp \qquad (8)$$

for the out-of-plane component of the magnetic fields[32]. Here, $D$ is the diffusion coefficient, and $\delta$ represents the thickness of the superconducting layer. We assume $\delta = 4.5$ Å, which is twice the height difference between the upper and lower atoms in the In bilayer (Fig. 1a). Note that $c_{O\parallel}$ is related to $c_{O\perp}$ as follows:

$$c_{O\parallel} = \frac{\pi\delta^2}{6\Phi_0} c_{O\perp}. \qquad (9)$$

The pair-breaking parameter for the paramagnetic effect in the presence of spin–orbit scattering is given by

$$\alpha_P = \frac{3\mu_B^2 B^2 \tau_{so}}{2\hbar} \equiv c_P B^2. \qquad (10)$$

The total pair-breaking parameter is the sum of all contributions and is given by Eq. (3). Near $T_{c0}$, the universal function Eq. (1) has an approximate form given by,

$$T_c = T_{c0} - \frac{\pi\alpha}{4k_B}. \qquad (11)$$

For the in-plane field, we take $B_\perp \simeq \theta_e B$ and $B_\parallel \simeq B$ in Eq. (3). The explicit form of the fitting function becomes,

$$T_c = T_{c0} - \frac{\pi}{4k_B}c_{O\perp}\theta_e B - \frac{\pi}{4k_B}(c_{O\parallel} + c_P)B^2. \qquad (12)$$

For the out-of-plane field, we take $B_\perp \simeq B$ and assumed that $c_{O\parallel}B_\parallel^2, c_P B_\parallel^2 \ll c_{O\perp}B_\perp$. The fitting function becomes,

$$T_c = T_{c0} - \frac{\pi}{4k_B}c_{O\perp}B. \tag{13}$$

These functions, (12) and (13), are fitted to the $T_c$ curves in Fig. 5b, d, respectively. $c_{O\parallel}$ and $c_P$ can be separated using Eq. (9). We confirmed that the estimated $\theta_e$ is within the accuracy in the sample angle control (see Supplementary Fig. 4).

## Data availability

The data that support the finding of this study are available from the corresponding author upon reasonable request.

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

## Acknowledgements

The authors thank Y. Higashi, S. Ichinokura and T. Shishidou for helpful discussions. We also thank K. Kuroda and A. Harasawa for their technical supports during the ARPES experiments. This work was supported financially by JSPS KAKENHI (Grant Numbers 18H01876, 16K17727, 25247053, 19H02592, 19H00651, 18K03484, 17H05211, and 17H05461), by Advanced Technology Institute (ATI) Research Grants 2017, and by World Premier International Research Center (WPI) Initiative on Materials Nanoarchitectonics, MEXT, Japan.

## Author contributions

S.Y. and T.U. conceived the experiment and wrote the manuscript. S.Y, K. Yokota, and T.U. carried out the electron transport experiments. S.Y. analysed the transport data and performed the DFT calculations. T.K., Y.N., K. Yaji, and K.S. measured the ARPES data with supports from F.K. and S.S. All the authors discussed the results and contributed to finalising the manuscript.

## Competing interests

The authors declare no competing interests.
