## [Peer Review File · Nature Communications]

Reviewers' comments:

Reviewer #1 (Remarks to the Author):

In this work, the authors studied the upper critical field (H_{c2}) of superconducting bilayer In on Si substrates. The authors argued that the Rashba spin-orbit coupling (SOC) in this material is strong but the Zeeman-type SOC is negligible. Theoretically, the Rashba SOC can enhance the H_{c2} only by a factor of $\sqrt{2}$ compared to the Pauli paramagnetic limit of superconductors with the same critical temperature without SOC. However, the authors claimed that the H_{c2} of the In thin film is three times as large as the Pauli limit, which is due to the spin-orbit scattering of the electrons.

The authors argued that the current system is analogous to the Zeeman-type SOC (or Ising SOC) protected superconductors which have been observed in many transition metal dichalcogenides but the enhancement is due to Rashba SOC instead of Zeeman-type SOC.

However, after reviewing the manuscript, I believe that this work does not meet the standard of Nature Communications. Here are the reasons:

1. The lack of novelty

One of the important reasons why superconductors with Zeeman-type SOC (or Ising SOC) had attracted so much attention both experimentally and theoretically in recent years is because Ising SOC provides a NEW mechanism to strongly enhance the in-plane H_{c2} of superconductors.

Before the discovery that Ising SOC can strongly enhance H_{c2} , it was found in many quasi two-dimensional superconductors (such as in intercalated transition metal dichalcogenides, please see Phys. Rev. B 21, 2717(1980)) that the H_{c2} could well exceed the Pauli limit. However, the enhancement of H_{c2} was attributed to spin-orbit scatterings. The reason why spin-orbit scatterings can enhance H_{c2} is that the constant flipping of electron spins can weaken the paramagnetic depairing effect. However, this mechanism would require spin-flip scatterings and the enhancement is strong when the superconductor is highly disordered. On the other hand, Ising SOC provided a mechanism to strongly enhance H_{c2} without any spin-flip scatterings.

In this work, the authors observed that the in-plane H_{c2} is enhanced. By extrapolation, they claimed that the H_{c2} is three times as large as the Pauli limit. As it is well known that Rashba SOC can enhance the H_{c2} only by a factor of $\sqrt{2}$ compared to the Pauli limit, the authors claimed that the enhancement is due to spin-flip scatterings. Indeed, I agree with the claim of the authors. But this kind of spin-flip scattering enhanced H_{c2} has been studied for decades [such as in Phys. Rev. B 21, 2717(1980)].

Moreover, Eq.1 of the manuscript, which was used to analyze the data, can be derived from a Hamiltonian with spin-flip scatterings. It has nothing to do with Rashba SOC [please see Eq.40 of Phys. Rev. B 12, 877 (1975)].

2. The claim is too strong given the lack of experimental data

It is well known that pure Rashba SOC without spin-flip scatterings can enhance the H_{c2} by a factor of $\sqrt{2}$. It is true that in Fig.4e, there are signs that the H_{c2} can be rather high. However, the authors only measured H_{c2} up to the Pauli limit at 5 Tesla. In recent experiments on Ising superconductors, the H_{c2} was measured above 30 Tesla and up to five times the Pauli limit in Ref. 8 [Science 350, 1353 (2015)]. In Ref. 9 [Nat. Phys. 12,144 (2016)], the H_{c2} was measured up to 55 Tesla and several times larger than the Pauli limit. Even in the work of Phys. Rev. B 21, 2717(1980) which was

performed more than three decades ago, the H_{c2} was measured up to 12 Tesla and above the Pauli limit.

Performing measurements only up to the Pauli limit and claiming that the H_{c2} can be three times the Pauli limit is not acceptable, at least not for a journal at the level of Nature Communications.

3. In the discussion, the authors mentioned that the current Rashba SOC may have the potential to be used to create chiral topological superconductors and the authors cited Ref.37 as a reference. I believe this statement is incorrect.

To create a chiral topological superconductor as pointed out in Ref.37, one needs to open a large Zeeman gap at some time-reversal invariant points and tune the chemical potential to be within the Zeeman gap. With the current system, the Fermi energy is 200meV above the band bottom at the Gamma point. It will be extremely difficult to open a large Zeeman gap, tune the chemical potential to the Zeeman gap and expect the intrinsic superconductivity to survive.

Moreover, in this system, there are many other bands with small Rashba SOC. When a large Zeeman field is coupled to these bands, these bands will remain gapless in the bulk even though the bands with large Rashba SOC can have superconducting pairing. These trivial gapless modes in the bulk will make the system topologically trivial. Therefore, in my opinion, the bands with very small Rashba SOC will make this system not desirable for creating chiral topological superconductors.

4. It is hard for the authors to rule out the effects of Ising SOC

In the theoretical band structure calculations, it is reasonable to expect Rashba SOC to be much stronger than Ising SOC, even when the In is coupled to the Si substrate. However, in realistic experiments, coupling to Si breaks many of the crystal symmetries of In. Due to the lattice mismatch, the lattice distortion can also break some mirror symmetries and induce Ising SOC. When a few meV of Ising SOC is induced on bands with small Rashba SOC, the H_{c2} can be strongly enhanced as well. It is very hard to rule out the Ising SOC effect experimentally.

Overall, I believe this work does not meet the standard of Nature Communications and I do not recommend its publication at this journal.

Reviewer #2 (Remarks to the Author):

The authors present a beautiful experiment to reveal the effect of Rashba-type SOC on superconductivity. The data is solid. The transport measurement in UHV is challenging, but has been well carried out by the authors. Here is my comment:

The work wants to prove that the dynamic spin-momentum locking is the mechanism for the large enhancement of upper critical field. The argument largely depends the theoretical analysis. c_P is determined by fitting T_c vs B with theory. Then the spin scattering time, which is much larger than that caused by SOC, is estimated from c_P . However, it is still not convincing why this enhanced scattering is due to the scattering on non-magnetic impurities. Are there other possibilities? A control experiment is needed to elucidate the role of impurity scattering. For example, under different sample preparation conditions, the density of impurities can be different. I would recommend the publication of the manuscript after the improvement.

The manuscript entitled “Atomic-layer Rashba-type superconductor protected by dynamic spin-momentum locking” reports a comprehensive study on the ultra thin indium films grown on atomically flat silicon substrates. The characterization includes DFT calculation, STM, ARPES, and electrical transport. From ARPES and DFT calculation, the authors are able to identify that the electronic bands splitting has a Rashba type of spin texture. The transport measurement under B field shows enhanced B_{c2} for the B_{\parallel} configuration compared with the theoretical prediction for the Rashba type band, $\sqrt{2}B_p$. The anomalously large H_{c2} in the B_{\parallel} configuration is then attributed to the Rashba type SOC in the bands across the E_F .

First of all, the data from the authors is of high quality. Especially, the consistency between the DFT calculation and ARPES is remarkable. Nevertheless, as the main claim is about the large upper critical field and its origin, then my focus is not on their impressive data but the plausibility of the underlying mechanism.

The argument is the following, I am fully convinced about a Rashba type spin-orbit splitting in the system. Also, there is a clear increase of the upper critical field in B_{\parallel} configuration that surpasses the B_p for 3 times. The main issue is then whether the Rashba SOC is the main contributor to the enhanced H_{c2} .

The enhancement of H_{c2} specifically in 2D has been studied theoretically in the KLB model (Ref. 1 listed below) long time ago. The H_{c2} can be significantly enhanced also due to the spin-orbit scattering. Note that in the KLB theory, Rashba spin-orbit scattering is not included. Therefore, in spite of the absence of Rashba splitting, it is still possible to have a large increase in the H_{c2} . Therefore, we need a clear argument to clarify why the Rashba splitting should be regarded as the main origin of the large H_{c2} . Namely, the spin-orbit scattering in the electronic bands of In film without Rashba SOC is still possible to give you the amount of enhancement according to the KLB theory. Also, as shown in Fig. 3a, Rashba type of splitting is not for all the bands across the Fermi level. There are bands without splitting as well.

To clarify this point, a lower T or high B experiment is then essential because the inclusion of Rashba SOC in the B_{\parallel} configuration is expected to show a clear dip (Fig. 4d in Ref. 2 listed below) in the $H_{c2}(T)$ dependence due to the finite partial coupling between the Rashba type of SOC with the B_{\parallel} . As shown in Fig. 1a, for the B_{\parallel} along k_x , the spin state with $k_y = 0$ are orthogonal to the B_{\parallel} , which is not affected by B field. However, the spin state with $k_x = 0$ is parallel to the B field. Therefore, Zeeman effect is not zero. Therefore, we are expecting to see both coupling and protection in their relevant energy scales as a function of temperature. Towards low temperature, if Rashba coupling is indeed there, we would observe the dip in B_{c2} as the hallmark of the Rashba type of coupling in the parallel field configuration.

The fitting to the $B_{c2}(T)$ also has quite a few free parameters, which can give sufficient freedom for having a consistent fitting in Fig. 5e. But for Flat #1 and Flat #2, the $B_{c2}(0)$ is very different. The reason is not clear to me. The different fitting parameters for the same type of Flat samples need justification.

Given the present evidence, the conclusion that the Rashba SOC is the cause of the large B_{c2} is premature. And we need the support of more clear-cut evidence.

References

1. Theory of the upper critical field in layered superconductors. PRB 12, 877–891 (1975).
2. Liu, Y. et al. Interface-Induced Zeeman-Protected Superconductivity in Ultrathin Crystalline Lead Films. Phys. Rev. X 8, 021002 (2018).

Authors:

We sincerely thank all the reviewers for their careful reading of our manuscript and for the number of insightful comments, which helped us to improve the quality of our work extensively. We are also grateful for the reviewers' positive comments such as "beautiful experiment", "The data is solid", "challenging, but has been well carried out", "of high quality", and "the consistency between the DFT calculation and ARPES is remarkable". In the following, we address the reviewers' issues point by point.

Reviewer #1:

1. The lack of novelty

One of the important reasons why superconductors with Zeeman-type SOC (or Ising SOC) had attracted so much attention both experimentally and theoretically in recent years is because Ising SOC provides a NEW mechanism to strongly enhance the in-plane H_{c2} of superconductors.

Before the discovery that Ising SOC can strongly enhance H_{c2} , it was found in many quasi two-dimensional superconductors (such as in intercalated transition metal dichalcogenides, please see Phys. Rev. B 21, 2717(1980)) that the H_{c2} could well exceed the Pauli limit. However, the enhancement of H_{c2} was attributed to spin-orbit scatterings. The reason why spin-orbit scatterings can enhance H_{c2} is that the constant flipping of electron spins can weaken the paramagnetic depairing effect. However, this mechanism would require spin-flip scatterings and the enhancement is strong when the superconductor is highly disordered. On the other hand, Ising SOC provided a mechanism to strongly enhance H_{c2} without any spin-flip scatterings.

In this work, the authors observed that the in-plane H_{c2} is enhanced. By extrapolation, they claimed that the H_{c2} is three times as large as the Pauli limit. As it is well known that Rashba SOC can enhance the H_{c2} only by a factor of $\sqrt{2}$ compared to the Pauli limit, the authors claimed that the enhancement is due to spin-flip scatterings. Indeed, I agree with the claim of the authors. But this kind of spin-flip scattering enhanced H_{c2} has been studied for decades [such as in Phys. Rev. B 21, 2717(1980)].

Moreover, Eq.1 of the manuscript, which was used to analyze the data, can be derived from a Hamiltonian with spin-flip scatterings. It has nothing to do with Rashba SOC [please see Eq.40 of Phys. Rev. B 12, 877 (1975)].

Authors:

We apologise that our description of the new mechanism was not clear enough. It is also our mistake that the important Klem-Luther-and Beasley (KLB) theory and a related experiment, which are highly relevant to this work, were not mentioned in the paper. We believe that our study contains important results because it proves the new mechanism of strong enhancement of H_{c2} in 2D superconductors with Rashba-type SOC.

First of all, we would like to point out that our bilayer In samples are highly crystalline as the recently studied TMDCs and they should be distinct from strongly disordered superconducting films studied for decades. As presented in Table 1 in the manuscript, the typical normal-state sheet

resistance of our samples is less than 100 Ohms. This value is two orders of magnitude smaller than the inverse of conductance quantum at which superconductor-insulator transition takes place. Moreover, the corresponding total elastic scattering time is several tens of femtoseconds. This time scale is comparable or even larger than those of the TMDC samples used in the studies of Zeeman-type SOC. (For example, total elastic scattering time is estimated to be 15-185 fs in Science 250, 1353 (2015) and 25.5-59.3 fs in Nat. Phys 12, 144 (2016)). This fact means that the degree of disorder of our samples is at the same level as the TMDC samples.

As the reviewer commented, it has been widely known that spin-orbit scattering can enhance H_{c2} by decreasing the paramagnetic depairing effect. But we note that this mechanism is relevant only to *disordered* materials because the spin-orbit scatterings occur much less frequently compared with total scattering events. In the present case, the spin-flip time τ_s should be longer by a factor of $60 - 10^3$ than the total elastic scattering time τ_{el} as described in the text. As our samples have a long τ_{el} due to their high crystallinity, the conventional mechanism can explain only a small fraction of the large H_{c2} enhancement factor. Thus we need to find a new mechanism to explain our results.

For this purpose, we adopted the proposal by Nam et al. [Proc. Natl. Acad. Sci. 113, 10513 (2016)], in which elastic scattering of electron populating spin-momentum locked bands with Rashba-type SOC cause an effective spin flipping. If this is the case, the spin-scattering time (τ_s) should be nearly equal to the elastic scattering time (τ_{el}). Indeed, this is exactly what we have found through our analysis; τ_s falls equal to τ_{el} within the experimental errors for the flat samples. For the vicinal samples, $\tau_s \sim 2\tau_{el}$ is found, but this discrepancy can be explained by the partial unlocking of the spin-momentum locking, which originates from the energy broadening of the Rashba-split bands as described in the text. This further strengthens the validity of the present argument. We thereby experimentally showed that the elastic scattering in spin-momentum locked bands due to Rashba-type SOC works analogously to the conventional spin-orbit scattering and gives rise to a strong suppression of paramagnetic pair-breaking effect.

We note here that the study by Nam et al. did not directly detect this phenomenon, because, in their few-layer Pb film, the orbital pair-breaking effect was still stronger than the paramagnetic pair-breaking effect even under in-plane magnetic fields. Instead, we used thinner crystalline superconducting films to successfully realise the condition in which the paramagnetic pair-breaking effect dominates the orbital pair-breaking effect under in-plane magnetic field.

To summarise, we have experimentally found a new mechanism for the strong enhancement of H_{c2} that is applicable to *crystalline* 2D superconductors. The essence of the finding is that electron scattering effectively plays the role of spin scattering under the influence of the Rashba-type SOC, which we refer to as dynamic spin-momentum locking. This is very different from the static protection mechanism due to the Zeeman-type SOC, or the conventional spin-orbit scattering. Since it does not exist without the Rashba-type SOC, it can be safely called a new mechanism of large H_{c2} enhancement.

Actions:

In the introduction, we have described that the conventional spin-orbit scattering enhances the H_{c2} of disordered superconductors, while dynamic spin-momentum locking can work in crystalline superconductors. Hereafter, ~~the removed part of the previous version is coloured blue and a strikethrough~~, while ~~the inserted part in the revised version is coloured red~~.

(Introduction, p.3)

The mechanism, referred to as dynamic spin-momentum locking here, ~~should dramatically enhance the spin-scattering rate~~ while preserving the time-reversal symmetry. This enhances $B_{c2||}$ through the suppression of paramagnetic pair-breaking effect.

was changed to

The mechanism, referred to as dynamic spin-momentum locking here, should cause **frequent spin scatterings** while preserving the time-reversal symmetry. This enhances $B_{c2||}$ through the suppression of paramagnetic pair-breaking effect **even in crystalline systems in an analogous manner as the conventional spin-orbit scattering does in disordered systems.**

In the results section, we have mentioned the fact that the R_n values of our samples are comparable to those of TMDC samples.

(Results, p.5)

The obtained parameters for T_{c0} and R_n are presented in Table 1. The small R_n of 36-90 Ω reflects the high crystallinity of the samples. **These values are comparable to those reported for transition-metal dichalcogenide samples used in the studies of Zeeman-type SOC [8, 9].**

The description of the fitting analysis was modified to highlight our logic more clearly. We have mentioned the assumptions made in the analysis in the beginning.

(Results, p.6)

It is widely known that spin scattering is induced occasionally at an elastic electron scattering event by the atomistic SOC. In the presence of this conventional spin-orbit scattering, Cooper pairs are no longer exact spin-singlet states.

...

Here we **assume this mechanism and, without taking account of the Rashba-type SOC,** analyse the magnetic field effects on superconductivity in terms of pair-breaking parameters.

Then, to clarify the aim of the analysis, we have modified the construction of the description about the evaluation process of spin and elastic scattering times. Specifically,

(Results, pp.7-8)

~~The above finding indicates that there exists a mechanism for inducing frequent spin scatterings. While a spin scattering may be caused by SOC at an elastic electron scattering even in a spin-degenerate system, this effect is weak in the present case. The ratio of elastic scattering time to spin scattering time τ_{el}/τ_s , a measure of the strength of SOC, is in the order of $(Z\alpha)^4$, where Z is the atomic number and $\alpha = 1/137$ the fine structure constant [29]. For In, $Z = 49$ gives $\tau_{el}/\tau_s \approx 1/60$. In the following, we show that spin scattering is strongly enhanced by dynamic spin-momentum locking and τ_{el}/τ_s approaches unity.~~ We first estimate elastic scattering time τ_{el} from the normal-state sheet resistance R_n .

(evaluation of τ_{el} and τ_s)

These results lead to $\tau_{el}/\tau_s \approx 0.5-1$, ~~indicating that spin scattering rate (i.e. inverse of τ_s) is strongly enhanced by a factor of 30–60 from the above estimation.~~

was changed to

The strong enhancement of $B_{c2||}$ observed above is actually not attributed to the atomistic SOC, but to the Rashba-type SOC as explained in the following. We first estimate elastic scattering time τ_{el} from the normal-state sheet resistance R_n .

(evaluation of τ_{el} and τ_s)

These results lead to $\tau_{el}/\tau_s \approx 0.5-1$. Nevertheless, if only the conventional spin-orbit scattering is considered, τ_s should be much larger than the τ_{el} . In this case, the ratio τ_{el}/τ_s should be on the order of $(Z\alpha)^4$, where Z is the atomic number and α is the fine structure constant [30]. For In ($Z = 49$), $\tau_{el}/\tau_s \sim 1/60$. An experimental study reported an even smaller τ_{el}/τ_s of about 10^{-3} for thin In films [37]. In contrast, if dynamic spin-momentum locking occurs on Rashba-split Fermi surfaces, we expect τ_{el}/τ_s approaching unity because every elastic scattering should contribute a spin flipping. This is consistent with our result of $\tau_{el}/\tau_s \approx 0.5-1$. The decrease in τ_s together with Eq. (3) and Eq. (6) means the paramagnetic pair breaking parameter α_p is suppressed by orders of magnitude from the value expected for the conventional spin-orbit scattering.

Here, we have cited a reference [37] reporting an experimental τ_{el}/τ_s value of about 10^{-3} for disordered In thin films. Accordingly, we changed the expression on the suppression in the pair-breaking parameter. Specifically,

(Abstract, p.2)

suppresses the Cooper pair-breaking parameter by ~~a factor of 30–60~~ and thereby protects superconductivity.

was changed to

suppresses the Cooper pair-breaking parameter by **orders of magnitude** and thereby protects superconductivity.

and similarly

(Introduction, p.4)

Our quantitative data analysis clarifies that the paramagnetic pair-breaking parameter α_p is strongly suppressed by ~~a factor of 30–60~~ from the value estimated for ~~a spin-degenerated Fermi surface~~.

was changed to

Our quantitative data analysis clarifies that the paramagnetic pair-breaking parameter α_p is strongly suppressed by **orders of magnitudes** from the value estimated for **the conventional spin-orbit scattering**.

We have also cited the following literature because they are highly relevant to the present work.

(Results, p.7)

This form of the pair-breaking parameter is closely related to the Klemm-Luther-Beasley (KLB) model proposed for 2D superconductors with conventional spin-orbit scattering [33, 34].

[33] Klemm, R. A., Luther, A. & Beasley, M. R. Theory of the upper critical field in layered superconductors. *Phys. Rev. B* 12, 877–891 (1975).

[34] Prober, D. E., Schwall, R. E. & Beasley, M. R. Upper critical fields and reduced dimensionality of the superconducting layered compounds. *Phys. Rev. B* 21, 2717–2733 (1980).

Finally, we have stated that our conclusion is indeed related to Rashba-type SOC.

(Results, p.8)

These findings and arguments provide compelling evidence that the dynamic spin-momentum locking is the dominant cause for anomalous suppression of paramagnetic pair breaking effect.

These findings and arguments provide compelling evidence that the dynamic spin-momentum locking **originating from Rashba-type SOC** is the dominant cause for anomalous suppression of paramagnetic pair breaking effect.

Reviewer #1:

2. The claim is too strong given the lack of experimental data

It is well known that pure Rashba SOC without spin-flip scatterings can enhance the H_{c2} by a factor of $\sqrt{2}$. It is true that in Fig.4e, there are signs that the H_{c2} can be rather high. However, the authors only measured H_{c2} up to the Pauli limit at 5 Tesla. In recent experiments on Ising superconductors, the H_{c2} was measured above 30 Tesla and up to five times the Pauli limit in Ref. 8 [*Science* 350, 1353 (2015)]. In Ref. 9 [*Nat. Phys.* 12,144 (2016)], the H_{c2} was measured up to 55 Tesla and several times larger than the Pauli limit. Even in the work of *Phys. Rev. B* 21, 2717(1980) which was performed more than three decades ago, the H_{c2} was measured up to 12 Tesla and above the Pauli limit.

Performing measurements only up to the Pauli limit and claiming that the H_{c2} can be three times the Pauli limit is not acceptable, at least not for a journal at the level of *Nature Communications*.

Authors:

We thank the referee for pointing out the critical issue. According to the referee's suggestion, we have performed an additional experiment by replacing the superconducting magnet of the cryostat with a larger one. In the revised manuscript, we have added data measured in magnetic fields up to 8.25 T. The field reached $\sqrt{2}$ times the Pauli limit, and we confirmed the survival of the superconductivity. This should clear the reviewer's concern about the validity of our claim.

It is reasonable that the referee has an impression that the magnetic fields used in our study are not high enough compared to those in related works. Actually, the past experiments on 2D superconductors have applied very high magnetic fields because they can use an existing or a commercially-available apparatus. This is not possible for our experiments, because our 2D

superconductors consist of atomic layers exposed on the surfaces, which can be easily destroyed by exposure to air or by a protection layer. (In the TMDC samples, the 2D conduction layer exists in the subsurface region rather than at the surface top layer.) To avoid sample degradation, we have to keep an ultra-high vacuum (UHV) condition from sample growth to cryogenic measurements down to temperatures less than 1 K. Moreover, for the current study, we need to apply strong magnetic fields and control their direction with 0.1-degree precision. We achieved these requirements for the first time by developing a special cryogenic apparatus compatible with UHV. The present manuscript contains the very first example of such data. Furthermore, thanks to the nature of the surface-exposed 2D system, we could directly compare the transport data with the electronic band structures revealed by ARPES. This can be done only with our apparatus. These technically new features should attract much attention of readers of *Nature Communications*.

Actions:

In the revised manuscript, we have included two additional sets of data taken on Flat#3 and Vicinal#3. The experiment on Flat#3 was done at high magnetic fields up to 8.25 T as the additional experiment required by the referee. The data of Vicinal#3 was obtained before the upgrade of our magnet system and was analysed to confirm the reproducibility. These additional results were presented in Table 1 and Figure 5, as well as Supplementary Information.

We have mentioned that the upper critical field of Flat#3 reached $\sqrt{2}$ times the Pauli limit.

(Results, p.7)

We note that this enhancement factor exceeds the value of $\sqrt{2}$, which is expected for the static effect of Rashba-type spin momentum locking. **This claim is directly evidenced by the maximum value of $B_{c2||} / B_{\text{Pauli}} = 1.43$ obtained for Flat#3.**

We have also stated that we cannot use either conventional cryogenic systems or high-magnetic-field systems for our crystalline In films.

(Results, p.4)

Despite these ideal features, studying superconducting properties of $\sqrt{7} \times \sqrt{3}$ -In is challenging because the susceptibility to foreign molecules and surface defects prohibits air exposure and the usage of ~~conventional cryogenic probes~~ [25, 26].

Despite these ideal features, studying superconducting properties of $\sqrt{7} \times \sqrt{3}$ -In is challenging because the susceptibility to foreign molecules and surface defects prohibits air exposure and the usage of **conventional cryogenic and high-magnetic-field systems** [25, 26].

Reviewer #1:

3. In the discussion, the authors mentioned that the current Rashba SOC may have the potential to be used to create chiral topological superconductors and the authors cited Ref.37 as a reference. I believe this statement is incorrect.

To create a chiral topological superconductor as pointed out in Ref.37, one needs to open a large Zeeman gap at some time-reversal invariant points and tune the chemical potential to be within the

Zeeman gap. With the current system, the Fermi energy is 200meV above the band bottom at the Gamma point. It will be extremely difficult to open a large Zeeman gap, tune the chemical potential to the Zeeman gap and expect the intrinsic superconductivity to survive.

Moreover, in this system, there are many other bands with small Rashba SOC. When a large Zeeman field is coupled to these bands, these bands will remain gapless in the bulk even though the bands with large Rashba SOC can have superconducting pairing. These trivial gapless modes in the bulk will make the system topologically trivial. Therefore, in my opinion, the bands with very small Rashba SOC will make this system not desirable for creating chiral topological superconductors.

Authors:

It is reasonable for the reviewer to have such concerns, but we do not claim that the present 2D superconductor $\sqrt{7}\times\sqrt{3}$ -In is a candidate for a topological superconductor. Rather, we are making a general argument on how the newly obtained insight helps to realise a topological superconductor, if some 2D superconductor can meet the requirement suggested by Referee. Even if this material is a candidate, the survival of superconductivity under strong magnetic field or exchange interaction is mandatory for the proposed model (Ref.37), which is generally difficult to realise. However, our finding of the new mechanism for enhanced H_{c2} may help to fulfil this condition, and thus can contribute to the study of topological superconductors. Furthermore, a multiband character with “trivial gapless mode” is not a serious problem for realising a topological superconductor. As theoretically clarified, the material becomes topological if the number of spin-split Fermi surfaces is odd [A. Y. Kitaev, Phys.-Usp. 44, 131 (2001)]. This issue has also been discussed for 2D systems more recently [for example, E. Dumitrescu et al., Phys. Rev. B 85 (2012)]. This fact alleviates the stringent condition for realising a topological superconductor and thus makes our argument more realistic.

Actions:

To clarify our claim, we have modified the discussion on the relation to topological superconductor as follows.

(Discussion, p.9)

The present result has significant implications in terms of robustness of a superconductor **with the Rashba-type SOC in general** under a strong magnetic field as well as in the proximity of a ferromagnet.

We have also added two references

(Discussion, p.9)

This makes realistic the coexistence of a 2D superconductor and a ferromagnet at atomic scales, which has been proposed to realise emergent phenomena such as chiral topological superconductivity [42–44].

[43] Kitaev, A. Y. Unpaired Majorana fermions in quantum wires. Phys.-Usp. 44, 131–136 (2001).

[44] Dumitrescu, E., Zhang, C., Marinescu, D. C. & Tewari, S. Topological thermoelectric effects in spin-orbit coupled electron- and hole-doped semiconductors. Phys. Rev. B 85 (2012).

Reference [42] corresponds to [37] in the previous version.

Reviewer #1:

4. It is hard for the authors to rule out the effects of Ising SOC

In the theoretical band structure calculations, it is reasonable to expect Rashba SOC to be much stronger than Ising SOC, even when the In is coupled to the Si substrate. However, in realistic experiments, coupling to Si breaks many of the crystal symmetries of In. Due to the lattice mismatch, the lattice distortion can also break some mirror symmetries and induce Ising SOC. When a few meV of Ising SOC is induced on bands with small Rashba SOC, the H_{c2} can be strongly enhanced as well. It is very hard to rule out the Ising SOC effect experimentally.

Authors:

The referee's comment that the coupling of the In layer to Si substrate breaks many of the crystal symmetries of bulk In is totally right. Actually, this effect has already been incorporated in the crystal structure model used in our density functional theory (DFT) calculations. The obtained structure is distorted from the free-standing structure, reflecting the coupling with the Si substrate. As a consequence of the breaking of the in-plane mirror symmetry, we see from Fig. 3d in the manuscript that a small portion of the states on the Fermi surface has Z components of the spin, as referee suggested.

This effect is, however, irrelevant in the presence of impurity scattering. As stated above, our system is characterised by dominant Rashba-type SOC. The polarisation axis of the spins strongly depends on momentum, which means that electron scattering between different momenta effectively induces a spin flipping. Although some of the states on the Fermi surface have a spin Z component, it is not conserved through elastic scattering. Thus, we cannot anticipate its major role in the enhancement of H_{c2} .

We note that a different situation takes place for an ultrathin layer of TMDC with Zeeman-type SOC. Here, since all the spins around one of the valleys are polarised in the same direction along the Z-axis, dominant intravalley scatterings cannot cause spin flipping. (The *intervalley* scattering can flip the spins, but this effect is very weak [Jose H. Garcia et al. Chem. Soc. Rev. 47, 3359 (2018)]. This is the reason why H_{c2} is enhanced by the protection of Zeeman-type SOC. By contrast, this mechanism does not apply to our Rashba-type SOC case because of the presence of frequent spin flipping.

Actions:

To clarify this point discussed above, we added the following statements in the revised manuscript.

(Results, p.5)

The effect of Zeeman-type SOC is negligible, judging from the fact that the out-of-plane components of spins are nearly absent (Fig. 3d). **This point will be discussed later in detail.**

(Discussion, p.9)

Another issue to be discussed is the possible effect of a finite Zeeman-type SOC, which is suggested from the non-zero out-of-plane spin polarisations shown in Fig. 3d. This effect is, however, irrelevant in the presence of elastic electron scattering. Since our system is characterised by dominant Rashba-type SOC, the polarisation axis of the spins strongly depends on momentum. This means that electron scattering between different momenta effectively induce a spin flipping and therefore out-of-plane spin component is not conserved through elastic scattering. Thus, we cannot anticipate its major role in the enhancement of $B_{c2\parallel}$. We note that a different situation takes place for an ultrathin layer of TMDC with Zeeman-type SOC [8, 9]. Here, since all the spins around one of the valleys are polarised in the same out-of-plane direction, dominant intravalley scatterings cannot cause spin flipping [41]. By contrast, this mechanism does not apply to our Rashba-type SOC case because of the presence of frequent spin flipping.

[41] Ilić, S., Meyer, J. S. & Houzet, M. Enhancement of the upper critical field in disordered transition metal dichalcogenide monolayers. *Phys. Rev. Lett.* 119 (2017)

Reviewer #2:

The work wants to prove that the dynamic spin-momentum locking is the mechanism for the large enhancement of upper critical field. The argument largely depends the theoretical analysis. c_P is determined by fitting T_c vs B with theory. Then the spin scattering time, which is much larger than that caused by SOC, is estimated from c_P . However, it is still not convincing why this enhanced scattering is due to the scattering on non-magnetic impurities. Are there other possibilities? A control experiment is needed to elucidate the role of impurity scattering. For example, under different sample preparation conditions, the density of impurities can be different. I would recommend the publication of the manuscript after the improvement.

Authors:

First of all, we apologise that our terminology was confusing. “Impurity” used here includes lattice defects such as atomic steps as well as impurity atoms because we do not expect distinct roles depending on the type of the scattering source in the present context. To clarify this point, we replace the term “non-magnetic impurity” with “non-magnetic scattering centre” in the caption of Figure 1.

With the above definition in mind, we have successfully controlled the density of scattering sources (“impurities”) by changing the miscut angle of the substrate; a larger miscut angle corresponds to a higher density of scattering sources. The density of scattering centres was evaluated from the normal-state sheet resistance. Interestingly, we have found that a high density of scattering sources (atomic steps) of the vicinal samples results in less effective spin scattering rate ($\tau_s \sim 2\tau_{el}$). This is because a high electron scattering rate causes an energy broadening of the spin-split Fermi surfaces and partially unlocks the spin-momentum relation. This supports our claim that spin-momentum locking is the cause of the effective spin flipping in the Rashba-type SOC system. Thus, we have carried out “control experiments” by changing the miscut angle of the surface as the referee suggested.

Concerning other types of impurities, we can judge the absence of magnetic impurities from the fact that the smallness of the decrease in T_{c0} sheet with increasing the normal-state sheet resistance. This is in sharp contrast with the strong suppression of T_{c0} by deposition of magnetic impurities found our previous study [Ref. 25 in the manuscript]. We agree that tuning the sample preparation condition may help us to elucidate the role of impurity scattering further. However, this issue is beyond the scope of the current study and we would like it to be left for the future study. At least, we have performed “control experiments” with vicinal samples as stated above, which was found to support our argument.

Actions:

To avoid confusion, we have changed the expression in the caption of Fig. 1.

(Figure 1, p.18)

(c) When an electron at the initial state k_i is elastically scattered to k_f by a non-magnetic impurity (depicted by a purple ball), its spin is forced to rotate.

was changed to

(c) When an electron at the initial state k_i is elastically scattered to k_f by a non-magnetic scattering centre (depicted by a purple ball), its spin is forced to rotate.

We have also mentioned the purpose of the use of vicinal surfaces. Specifically,

(Results, p.5)

~~Four~~ $\sqrt{7} \times \sqrt{3}$ -In samples were prepared for electron transport experiments: ~~two with nominally flat Si(111) surfaces (Flat#1/#2) and two with vicinal surfaces (Vicinal#1/#2, miscut angle 0.5°).~~

was changed to

Six $\sqrt{7} \times \sqrt{3}$ -In samples were prepared for electron transport experiments. **In addition to three nominally flat Si(111) surfaces (Flat#1/#2/#3), we used three vicinal surfaces (Vicinal#1/#2 with a miscut angle of 0.5° and Vicinal#3 with a miscut angle of 1.1°) to control the density of scattering sources.**

Since we have included additional sets of data, other parts of the same sentence were changed accordingly.

Reviewer #3:

Nevertheless, as the main claim is about the large upper critical field and its origin, Then my focus is not on their impressive data but the plausibility of underlying mechanism.

Authors:

We apologise that our description of the mechanism of the enhanced critical field was not convincing enough. We have revised our manuscript according to the referee's comments.

Reviewer #3:

The argument is the following, I am fully convinced about a Rashba type spin-orbit splitting in the system. Also, there is a clear increase of the upper critical field in B // configuration that surpasses the B_p for 3 times. The main issue is then whether the Rashba SOC is the main contributor to the enhanced H_{c2} .

The enhancement of H_{c2} specifically in 2D has been studied theoretically in KLB model (Ref. 1 listed below) long time ago. The H_{c2} can be significantly enhanced also due to the spin-orbit scattering. Note that in the KLB theory, Rashba spin-orbit scattering is not included. Therefore, in spite of the absence of Rashba splitting, it is still possible to have a large increase in the H_{c2} . Therefore, we need a clear argument to clarify why the Rashba splitting should be regarded as the main origin of the large H_{c2} . Namely, the spin-orbit scattering in the electronic bands of In film without Rashba SOC is still possible to give you the amount of enhancement according to the KLB theory.

Authors:

We thank the referee for kindly reminding us of the important work by Klem-Luther-Beasley (KLB) that is closely related to our study. The KLB model describes the critical field of layered superconductors and that of purely two-dimensional superconductors by taking the limit of small inter-layer coupling. It deals with the spin-flip scattering coming from the spin-orbit coupled term in the scattering potential, as well as the orbital pair-breaking effect that depends on the magnetic field angle. In fact, the KLB model and the model used in our analysis are both based on the same physics. While the KLB model can deal a multilayer system but our model cannot, the latter includes the term describing the small orbital effect within the superconducting layer for the *in-plane* component of the magnetic field by adopting the formula given in the textbook by Tinkham. To clarify this background, we added a reference to KLB's paper.

It is true that spin-orbit scattering by Rashba-type SOC is not included in our model (or in the KLB model). Since the effect of the conventional spin-orbit scattering by the atomistic SOC on the H_{c2} is widely known, we first analysed the data based on it without taking account of the Rashba-type SOC. In this way, we deduced successfully spin scattering time τ_s and compared it elastic electron scattering time τ_{el} . The result was surprising; we found $\tau_{el} = \tau_s$ within the experimental error for the flat samples and $\tau_{el} \sim 0.5\tau_s$ for the vicinal samples. If only the conventional spin-orbit scattering mechanism is considered, $\tau_{el}/\tau_s \sim (\alpha Z)^4 = 1/60$ for In, where α is the fine structure constant and Z is the atomic number ($Z = 49$). An experimental study reported even larger τ_{el}/τ_s of about 10^{-3} for thin

In films with a thickness of a few tens of nm [Physics Letters A 58, 131 (1976)]. Therefore, the spin-orbit scattering that occurs in the absence of Rashba SOC cannot account for the $\tau_{el}/\tau_s \sim 0.5-1$ obtained from our experiment. Nevertheless, if Rashba SOC is considered, this experimental result can be reasonably explained based on the concept of dynamic spin-momentum locking. This clearly shows why the Rashba splitting should be regarded as the primary origin of the large H_{c2}

Actions:

To clarify the relation to the KLB theory, we have added the following description.

(Results, p.7)

This form of the pair-breaking parameter is closely related to the Klemm-Luther-Beasley (KLB) model proposed for 2D superconductors with conventional spin-orbit scattering [33, 34]. In the present study, the addition of the $\alpha_{0\parallel}$ term allows us to account for the orbital effect within the superconducting layer under the *in-plane* magnetic field, which is not included in the KLB model. This effect played a crucial role in few-layer Pb films studied previously [10].

[33] Klemm, R. A., Luther, A. & Beasley, M. R. Theory of the upper critical field in layered superconductors. Phys. Rev. B 12, 877–891 (1975).

[34] Prober, D. E., Schwall, R. E. & Beasley, M. R. Upper critical fields and reduced dimensionality of the superconducting layered compounds. Phys. Rev. B 21, 2717–2733 (1980).

Reviewer #3:

Also, as shown in Fig. 3a, Rashba type of splitting is not for all the bands across the Fermi level. There are bands without splitting as well.

Authors:

This issue is important, but we would like to remind the reviewer that the histogram of energy splitting in Fig. 3(c) shows how the bands are split on the whole Fermi surface. It shows that the portion of the Fermi surface plotted in dark colours in Fig. 3(a) have a finite splitting of the order of a few meV.

Reviewer #3:

To clarify this point, a lower T or high B experiment is then essential because the inclusion of Rashba SOC in the $B \parallel$ configuration is expected to show a clear dip (Fig. 4d in Ref. 2 listed below) in the $H_{c2}(T)$ dependence due to the finite partial coupling between the Rashba type of SOC with the $B \parallel$. As shown in Fig. 1a, for the $B \parallel$ along k_x , the spin state with $k_y = 0$ are orthogonal to the $B \parallel$, which is not affected by B field. However, the spin state with $k_x = 0$ is parallel to the B field. Therefore, Zeeman effect is not zero. Therefore, we are expecting to see both coupling and protection in their relevant energy scales as a function of temperature. Towards low temperature, if Rashba coupling is indeed there, we would observe the dip in H_{c2} as the Hallmark of the Rashba type of coupling in the parallel field configuration.

Authors:

We thank the referee for suggesting a way to test the role of Rashba-type SOC in the enhancement of the upper critical field. We would like to answer the referee's comment point by point in the following.

(1) Dip in the H_{c2} curve in the presence of both Zeeman- and Rashba-type SOCs

We do not think the presence of a dip in the H_{c2} curve proves the dominant role of the Rashba-type SOC. The H_{c2} curves given by the theoretical calculation in Phys. Rev. X 8, 021002 (2018) are characterised by two parameters α_R and β_{SO} representing the disorder-corrected magnitudes of the Rashba-type SOI and the Zeeman-type SOI, respectively. The dip mentioned by the referee is shown in Fig. 4d of the paper. It emerges when α_R is increased with β_{SO} kept constant and is present even when β_{SO} is decreased to 0. However, in other theoretical papers [Phys. Rev. B 76, 014522 (2007), Phys. Rev. B 78, 224520 (2008)], H_{c2} curves computed for purely Rashba-type SOI are monotonic functions of temperature and do not display any dip. This means that the presence of dip depends on the details of the theoretical model. For example, it has been pointed out that in the presence of Rashba-type SOC the superconducting order parameter exhibits a field-induced spatial modulation in the direction perpendicular to the in-plane magnetic field, which leads to a correction to the upper critical field by a factor of $[(v_F^2 + \alpha^2)/(v_F^2 - \alpha^2)]^{1/2}$, where v_F is the Fermi velocity [Phys. Rev. B 92, 014509 (2015)]. On the other hand, the description of the theoretical formulation in Phys. Rev. X 8, 021002 (2018) does not mention this correction, and the presence and absence of this correction may produce some differences in the details of the H_{c2} curves compared with other calculations.

(2) Protection and coupling of in-plane magnetic fields due to momentum-dependent spin orientation

The reviewer also pointed out that the different directions of the spins on the Fermi surface along $k_x = 0$ and $k_y = 0$ may give rise to both coupling and protection from the Zeeman effect under in-plane magnetic fields. This is totally true in a clean 2D superconductor with Rashba-type SOC. When the magnetic field is in-plane, some of the spins are protected against the magnetic field and some of the spins are coupled to the magnetic field. Consequently, the paramagnetic depairing effect is only partially suppressed, resulting in the factor $\sqrt{2}$ increase of the upper critical field from the Pauli limit. However, it is hard to account for the factor 3 increase observed in our experiment by this mechanism.

Reviewer #3:

The fitting to the $B_{c2}(T)$ also has quite a few free parameters, which can give sufficient freedom for having a consistent fitting in Fig. 5e. But for Flat #1 and Flat #2, the $B_{c2}(0)$ is very different. The reason is not clear to me. The different fitting parameters for the same type of Flat samples need justification.

Authors:

We are grateful that the reviewer thoroughly checked our data and has raised an important question regarding the reproducibility. The difference pointed by the reviewer is due to the out-of-plane component of the magnetic fields. Although we carefully adjusted the rotation angle of each sample with respect to the magnetic field in-situ by monitoring the resistance minimum at a precision of the order of 0.1 degrees, the adjustment was not perfect and small out-of-plane components may remain. Its magnitude differs by sample to sample and leads to a sample-dependent contribution to the pair-breaking parameter represented by $c_{O\perp B\perp}$ in Eq. (3). The fitting curves presented in Fig. 5e includes the contribution of this term. This is the reason why $B_{c2}(0)$ differs from sample to sample even for the Flat substrate.

Actions:

We have added the following explanation in the caption of Fig. 5 in the revised manuscript.

(Figure 5, p.24)

The relatively large variation in the fitting curves for $B_{c2\parallel}$ originates mainly from the angular error of the sample orientation denoted by θ_e in the main text.

Reviewer #3:

Given the present evidence, the conclusion that the Rashba SOC is the cause of the large B_{c2} is premature. And we need the support of more clear-cut evidence.

References

Theory of the upper critical field in layered superconductors. PRB 12, 877-891 (1975).

Liu, Y. et al. Interface-Induced Zeeman-Protected Superconductivity in Ultrathin Crystalline Lead Films. Phys. Rev. X 8, 021002 (2018).

Authors:

With the above argument and justification in mind, we are confident that the Rashba-type SOC is the cause of the large H_{c2} . We thank the referee for introducing important literature. Both papers are cited in the revised manuscript.

Summary of other changes:

(1) Addition of a new author and affiliation

The additional experiment required by Reviewer #1 was performed with Kenta Yokota, a new member of our group. We have added him to the author list in the revised manuscript.

Kenta Yokota,^{5,6}

⁵*International Center for Materials Nanoarchitectonics (WPI-MANA), National Institute for Materials Science, 1-1 Namiki, Tsukuba, Ibaraki 305-0044, Japan*

⁶*Department of Condensed Matter Physics, Graduate School of Science, Hokkaido University Kita 8, Nishi 5, Kita-ku, Sapporo, Hokkaido 060-0808, Japan*

We have added the latter affiliation to Takashi Uchihashi, who is the supervisor of Kenta Yokota.

Accordingly, the description in the Author Contributions has been changed as follows.

(Author Contributions, p.16)

S.Y. and T.U. conceived the experiment and wrote the manuscript. S.Y, K. Yokota, and T.U. carried out the electron transport experiments.

The official affiliation of Takahiro Kobayashi was changed in accordance with his supervisor Kazuyuki Sakamoto, who has been moved from Chiba University to Osaka University in advance.

We have also added the present address of Koichiro Yaji.

(2) Correction of errors

We have corrected a wrong reference to a figure.

(Results, p.7)

All coefficients were determined by fitting Eq. (1) to the experimental data in Figs. 5b and 5d,

→ All coefficients were determined by fitting Eq. (1) to the experimental data in Figs. 5c and 5d,

The coefficient of Eq. (10) was wrong. This is simply a typo, and the coefficient of the related equation [Eq. (6)] was correct. The results presented in the manuscript are not affected.

(Materials and Methods, p.11)

$$\alpha_P = \frac{4\mu_B^2 B^2 \tau_{SO}}{3\hbar} \equiv c_P B^2$$

was corrected as

$$\alpha_P = \frac{3\mu_B^2 B^2 \tau_{SO}}{2\hbar} \equiv c_P B^2$$

We have also corrected an error and inconsistency in the caption of Fig. 5 as follows.

(Figure 5, p.24)

The ~~vertical~~ line indicates the enhancement factor $\sqrt{2}$ for static locking effect of Rashba SOC.

→ The ~~dashed horizontal~~ line indicates the enhancement factor $\sqrt{2}$ for static locking effect of Rashba-type SOC.

(3) Correction regarding the additional experiment and data

In the revised manuscript, we have added results for Flat#3 and Vicinal#3 samples. Table 1 and Figure 5 now include additional data. Also, since the experiment on Flat#3 was done in the upgraded apparatus, we modified the description of the Electron transport subsection in the Materials and Methods. Specifically,

(Materials and Methods, p.10)

For transport experiments, ~~four~~ samples were grown on substrates cut from Si(111) wafers (3 mm × 8 mm × 0.38 mm) with miscut angles of 0° (Flat#1 and Flat#2) or 0.5° (Vicinal#1 and Vicinal#2) in the [-1 -12] direction.

was changed to

For transport experiments, ~~six~~ samples were grown on substrates cut from Si(111) wafers (3 mm × 8 mm × 0.38 mm) with miscut angles of 0° (Flat#1, Flat#2, and Flat#3), 0.5° (Vicinal#1 and Vicinal#2), and 1.1° (Vicinal#3) in the [-1 -12] direction.

and

(Materials and Methods, p.11)

The samples were then cooled down to ~0.9 K ~~by pumping condensed ⁴He with a charcoal sorption pump~~. The magnetic fields were applied with a ~~5-T solenoid magnet~~.

was changed to

The samples were then cooled down to ~0.9 K or to ~0.4 K by pumping condensed ⁴He or ³He with a charcoal sorption pump. The magnetic fields were applied with a superconducting solenoid magnet. The maximum field was 5 T in the experiment of Flat#1/#2 and Vicinal#1/#2/#3 and was 8.25 T in the experiment of Flat#3.

We also made several small changes that are all related to the additional data.

(Results, p.5)

The curves of the other ~~two~~ samples are available in Supplementary Figure 1.

→ The curves of the other **four** samples are available in Supplementary Figure 1.

(Results, p.6)

Figure 5c shows the magnetic field dependence of T_c of all ~~four~~ samples,

→ Figure 5c shows the magnetic field dependence of T_c of all **six** samples,

(Results, p.6)

The data show that the lowering of T_c as a function of B is quadratic and reaches ~~6–8% of T_{c0} at 5 T~~.

→ The data show that the lowering of T_c as a function of B is quadratic and reaches **23% of T_{c0} at 8.25 T for Flat#3**.

(Results, p.8)

For flat samples, $\hbar/\tau_{el} = 9\text{--}12$ meV $< \Delta_R$,

→ For flat samples, $\hbar/\tau_{el} = 9\text{--}14$ meV $< \Delta_R$,

(Table 1, p.17)

Table 1: List of parameters obtained for the ~~six~~ samples.

→ Table 1: List of parameters obtained for the **six** samples.

(Table 1, p.17)

The values in the parentheses are the estimates of errors propagated from the accuracy of the calibration curve for magnetoresistance of the temperature sensor (0.005 K) and the hysteresis of the superconducting magnet (0.004 T).

→ The values in the parentheses are the estimates of errors propagated from the accuracy of the calibration curve for magnetoresistance of the temperature sensor (0.005 K **for $B \leq 5$ T and 0.04 K for $B \geq 5$ T**) and the hysteresis of the superconducting magnet (0.004 T).

(4) Supplementary information

Supplementary Figures 1, 2, 3 now include the corresponding data of Flat#3 and Vicinal#3. We have also corrected the reference numbers in the Supplementary Notes 1 and 2 according to the changes in the main text.

(5) Data availability statement

We updated the data availability statement according to an example suggested by Springer Nature.

The data that support the finding of this study are available from the corresponding author upon reasonable request.

Reviewers' comments:

Reviewer #1 (Remarks to the Author):

After reading the manuscript and the reply, I still find many issues that are not resolved in a satisfactory way in the manuscript.

1. The main claim of the authors is that "Our quantitative analysis clarifies that dynamic spin-momentum locking, a mechanism where spin is forced to flip at every elastic electron scattering, suppresses the Cooper pair-breaking parameter by orders of magnitude and thereby protects superconductivity"

This claim was obtained by Eq.1 and Eq.6 of the manuscript. However, I would like to point out that the estimation of τ_s is NOT reliable when the system has spin-momentum locking.

τ_s in Eq.1 (through the dependence of C_p on τ_s) is an effective parameter which includes all enhancement effects of H_{c2} in Eq.1. It can be an estimation of the actual spin-orbit scattering time only if there are NO spin-orbit splitting in the band structure.

In the presence of Rashba spin-orbit coupling (SOC), even without any back scattering events, one can find a small τ_s using Eq.1. This is because in Eq.1, all enhancement effects of H_{c2} are attributed to τ_s . In the case of Ising superconductors, one can even conclude that τ_s can be smaller than the τ_{el} which is unphysical.

In the authors' estimation of τ_s , they indeed included both the enhancement from the spin-momentum locking due to Rashba SOC and possible spin-orbit scattering events (and other possible effects). As a result, the authors overestimated the effect of τ_s . In Table I, it is clear that τ_s is shorter than τ_{el} for samples Flat #2 and Flat #3. This is not physical as there would be more spin-flip scatterings than the elastic scatterings. They claimed that this is within the experimental error in the main text. I don't think this is the reason.

2. The authors claimed that "in-plane upper critical magnetic field is anomalously enhanced" in their system.

As pointed out in the previous report, "pure Rashba SOC without spin-flip scatterings can enhance the H_{c2} by a factor of $\sqrt{2}$." And spin-flip scatterings can enhance H_{c2} further. It has been known for a long time that even scalar potentials can enhance H_{c2} in noncentrosymmetric superconductors. For Rashba superconductors, it is already known that even scalar disorders can enhance H_{c2} beyond the factor of $\sqrt{2}$ enhancement. Please refer to Mineev and Samoklin, Physical Review B 75, 184529 (2007) for the theoretical calculations. The same results can also be found in the chapter "Effects of Impurities in Non-centrosymmetric Superconductors" by Samoklin in the book entitled "Non-Centrosymmetric Superconductors" by Bauer and Sigrist.

Therefore, the results found by the authors are not anomalous. Disorder can indeed enhance the H_{c2} of Rashba superconductors and this is a well-known result.

In the previous version of this work, the authors did not even mention the Klemm-Luther-Beasley theory. Apparently, the authors were not aware that disorder can generally enhance H_{c2} in noncentrosymmetric superconductors. Instead, the authors claimed that they discovered new results which will provide "a new insight into how superconductivity can survive the detrimental effects of strong magnetic fields". The claimed results have in fact been proposed many years ago.

3. Even though H_{c2} is enhanced in the double layer In with Rashba SOC and disorders, it does not mean that the enhancement is purely due to the disorder enhanced spin-flip scattering. I still believe that the authors cannot rule out the effects of Ising SOC.

The system certainly breaks the mirror symmetries with mirror planes perpendicular to the samples. Therefore, the Ising SOC terms can arise. Even a few meV of band splitting due to Ising SOC can further enhance H_{c2} . The authors claimed that the Rashba SOC is dominant but this is only true for a small portion of the bands at the Fermi energy. The Rashba SOC can be very small in some parts of the bands. It is extremely difficult for DFT calculations to capture the few meV of band splitting due to Ising SOC. The presence of the Ising SOC terms can strongly enhance H_{c2} . Using the observed H_{c2} to estimate τ_s would produce an incorrect result and could result in the situation where τ_s is smaller than τ_{el} as shown in Table 1.

Given the above points, I believe that this work does not match the standard of Nature Communications.

Reviewer #2 (Remarks to the Author):

The concerns have been properly addressed in the revised manuscript. It is acceptable for publication.

Reviewer #3 (Remarks to the Author):

I've read through the revised manuscript of NCOMMS-19-42030A, the answers to my questions, and also the replies to the comments from the other two referees. I felt that most of the points are addressed convincingly with sufficient improvements.

For the crucial point of spin-orbit scattering. I would suggest the authors to consider adding their discussion (rewrite it into a more concise form) below to the main text to highlight the essential role of Rashba SOC.

"It is true that spin-orbit scattering by Rashba-type SOC is not included in our model (or in the KLB model). Since the effect of the conventional spin-orbit scattering by the atomistic SOC on the H_{c2} is widely known, we first analysed the data based on it without taking account of the Rashba-type SOC. In this way, we deduced successfully spin scattering time τ_s and compared it elastic electron scattering time τ_{el} . The result was surprising; we found $\tau_{el} = \tau_s$ within the experimental error for the flat samples and $\tau_{el} \sim 0.5\tau_s$ for the vicinal samples. If only the conventional spin-orbit scattering mechanism is considered, $\tau_{el}/\tau_s \sim (\alpha Z)^4 = 1/60$ for In, where α is the fine structure constant and Z is the atomic number ($Z = 49$). An experimental study reported even larger τ_{el}/τ_s of about $10-3$ for thin In films with a thickness of a few tens of nm [Physics Letters A 58, 131 (1976)]. Therefore, the spin-orbit scattering that occurs in the absence of Rashba SOC cannot account for the $\tau_{el}/\tau_s \sim 0.5-1$ obtained from our experiment. Nevertheless, if Rashba SOC is considered, this experimental result can be reasonably explained based on the concept of dynamic spin-momentum locking. This clearly shows why the Rashba splitting should be regarded as the primary origin of the large H_{c2} ".

Authors:

We thank all the reviewers for carefully reading and evaluating our revised manuscript. We are pleased to find the very positive comments by Reviewers #2 and #3: "*It is acceptable for publication*", "*I've read through the revised manuscript of NCOMMS-19-42030A, the answers to my questions, and also the replies to the comments from the other two referees. I felt that most of the points are addressed convincingly with sufficient improvements.*" In the following, we would like to respond to the remaining issues point-by-point.

Reviewer #1 (Remarks to the Author):**Reviewer:**

1. The main claim of the authors is that "Our quantitative analysis clarifies that dynamic spin-momentum locking, a mechanism where spin is forced to flip at every elastic electron scattering, suppresses the Cooper pair-breaking parameter by orders of magnitude and thereby protects superconductivity"

This claim was obtained by Eq.1 and Eq.6 of the manuscript. However, I would like to point out that the estimation of τ_s is NOT reliable when the system has spin-momentum locking.

τ_s in Eq.1 (through the dependence of C_p on τ_s) is an effective parameter which includes all enhancement effects of H_{c2} in Eq.1. It can be an estimation of the actual spin-orbit scattering time only if there are NO spin-orbit splitting in the band structure.

In the presence of Rashba spin-orbit coupling (SOC), even without any back scattering events, one can find a small τ_s using Eq.1. This is because in Eq.1, all enhancement effects of H_{c2} are attributed to τ_s . In the case of Ising superconductors, one can even conclude that τ_s can be smaller than the τ_{el} which is unphysical.

In the authors' estimation of τ_s , they indeed included both the enhancement from the spin-momentum locking due to Rashba SOC and possible spin-orbit scattering events (and other possible effects). As a result, the authors overestimated the effect of τ_s . In Table I, it is clear that τ_s is shorter than τ_{el} for samples Flat #2 and Flat #3. This is not physical as there would be more spin-flip scatterings than the elastic scatterings. They claimed that this is within the experimental error in the main text. I don't think this is the reason.

Authors:

Concerning the estimation error of τ_s , we first point out that the effect of the Zeeman SOC on B_{c2} should be negligible as we discuss below in our response to Comment 3. (Note: we use the terms "Zeeman SOC" instead of "Ising SOC" and B_{c2} instead of H_{c2}). Therefore, the only possible cause for error is the *static* effect of the Rashba SOC, which can enhance B_{c2} up to $\sqrt{2}$ times the Pauli limit. This static effect is likely to be weakened by electron scattering and mixing between different spin states. Since we do not know the degree of such an effect, we assume the worst case and estimate the upper limit of error in τ_s in the following.

In the absence of spin-orbit coupling, the B dependence of T_c near T_{c0} is given by

$$1 - \frac{T_c}{T_{c0}} = \frac{7\zeta(3)}{4\pi^2} \cdot \frac{(\mu_B B)^2}{(k_B T_{c0})^2} \dots (1)$$

With the static spin-momentum locking due to Rashba SOC, the expression changes to

$$1 - \frac{T_c}{T_{c0}} = \frac{1}{2} \cdot \frac{7\zeta(3)}{4\pi^2} \cdot \frac{(\mu_B B)^2}{(k_B T_{c0})^2} \dots (2)$$

(See V. Barzykin and L. P. Gor'kov, Phys. Rev. Lett. 89, 227002 (2002).) Here the addition of the factor 1/2 in Eq.(2) means that B is replaced with an effective magnetic field $B_{\text{eff}} = (1/\sqrt{2})B$ in Eq.(1). This is the origin of the enhancement of B_{c2} by a factor of $\sqrt{2}$ due to the static locking effect of Rashba SOC. The effect of non-magnetic disorder on a Rashba superconductor can be estimated using this effective magnetic field. By substituting B with $B_{\text{eff}} = (1/\sqrt{2})B$ in the following equations (taken from Eqs.(3) and (6) in the manuscript; only the paramagnetic contribution is considered here),

$$\alpha(B) = c_p B^2 \dots (3)$$

$$c_p = 3\tau_s \mu_B^2 / 2\hbar \dots (4)$$

we see that τ_s value is doubled for the same experimental data $\alpha(B)$. With the τ_s values obtained previously (see Table 1) in the manuscript, we can estimate that the lower limit of τ_{el}/τ_s is 0.25-0.5. These values are still much higher than 1/60-1/1000 for thin In films, which is due to the atomistic spin-orbit scattering mechanism. Therefore, our result cannot be understood in terms of the conventional mechanism and our conclusion remains intact. Indeed, by taking into account the static effect of Rashba SOC, we can explain the reason why τ_s is shorter than τ_{el} for some of the samples (on the assumption that the experimental error is smaller than our estimation).

Actions:

New sentences, a note and a reference have been added as follows.

- main text

(p.8)

“This argument further supports our conclusion on the critical role of the dynamic effect of the Rashba-type SOC.

Finally, we note that the static spin-momentum locking due to the Rashba-type SOC can enhance....
.... and our conclusion remains the same.”

- supplementary file

Supplementary Note 3

Supplementary References 52

Reviewer:

2. The authors claimed that “in-plane upper critical magnetic field is anomalously enhanced” in their system.

As pointed out in the previous report, “pure Rashba SOC without spin-flip scatterings can enhance the H_{c2} by a factor of $\sqrt{2}$.” And spin-flip scatterings can enhance H_{c2} further. It has been known for a long time that even scalar potentials can enhance H_{c2} in noncentrosymmetric superconductors. For Rashba superconductors, it is already known that even scalar disorders can enhance H_{c2} beyond the factor of $\sqrt{2}$ enhancement. Please refer to Mineev and Samoklin, Physical Review B 75, 184529 (2007) for the theoretical calculations. The same results can also be found in the chapter “Effects of Impurities in Non-centrosymmetric Superconductors” by Samoklin in the book entitled “Non-Centrosymmetric Superconductors” by Bauer and Sigrist.

Therefore, the results found by the authors are not anomalous. Disorder can indeed enhance the H_{c2} of Rashba superconductors and this is a well-known result.

In the previous version of this work, the authors did not even mention the Klemm-Luther-Beasley theory. Apparently, the authors were not aware that disorder can generally enhance H_{c2} in noncentrosymmetric superconductors. Instead, the authors claimed that they discovered new results which will provide “a new insight into how superconductivity can survive the detrimental effects of strong magnetic fields”. The claimed results have in fact been proposed many years ago.

Authors:

First of all, we point out that the paper cited by the reviewer (V. P. Mineev and K. V. Samokhin, Phys. Rev. B 75, 184529 (2007)) deals with the orbital pair-breaking effect in non-centrosymmetric superconductors with the disorder. The paper is not relevant for our atomically thin superconductor under in-plane magnetic fields, where the paramagnetic pair-breaking effect plays a major role. Instead, from the first version of the manuscript, we have cited another Samokhin's paper including the paramagnetic effect (K. V. Samokhin, Phys. Rev. B 78, 224520 (2008)) as well as two other theoretical papers about the paramagnetic pair-breaking effect in non-centrosymmetric superconductors in the presence of non-magnetic disorder (O. Dimitrova and M. V. Feigel'man, Phys. Rev. B 76, 014522 (2007) and M. Houzet and J. S. Meyer, Phys. Rev. B 92, 014509 (2015)). We have also discussed them in the manuscript within the context of our result. Therefore, we had already known very well the theoretical studies on the non-magnetic disorder effects on Rashba-type superconductors even before the reviewer pointed it out.

The reviewer's comment “*the results found by the authors are not anomalous. Disorder can indeed enhance the H_{c2} of Rashba superconductors and this is a well-known result.*” does not address the issue in the present paper. Our main message is not the finding of the enhancement of the B_{c2} itself, but its mechanism where the dynamic spin-momentum locking plays the essential role. This mechanism was clarified for the first time based on the quantitative comparison of τ_s and τ_{el} deduced from our experiment. None of the previous theories on the disorder-induced B_{c2} enhancement have mentioned this mechanism so far.

We also point out that, while the disorder-induced B_{c2} enhancement in Rashba-type superconductors was theoretically predicted, it has been elusive from the experimental point of view. So far, non-centrosymmetric bulk superconductors such as CePt₃Si were used to investigate this effect, but it was difficult to obtain conclusive evidence. This is because they often suffer from a non-negligible orbital pair-breaking effect as well as strong electron correlation effects and the detailed structures of the Fermi surfaces cannot be observed directly. In fact, the Samokhin's chapter in the Bauer &

Sigrist book is concluded by enumerating various issues on the experimental studies on CePt3Si and stating that "*In order to resolve these issues, more systematic studies of the disorder effects in a wide range of impurity concentrations are needed.*" These facts mean that our finding is far from being "well-known result" as the reviewer claims. Our work offers the first conclusive experimental evidence on this effect. It has become possible because our atomic-layer material has a simple chemical composition and has well-defined crystal and band structures. This allows us to carry out detailed analysis based on the realistic Fermi surface structure and surface crystal structure.

Reviewer:

3. Even though H_{c2} is enhanced in the double layer In with Rashba SOC and disorders, it does not mean that the enhancement is purely due to the disorder enhanced spin-flip scattering. I still believe that the authors cannot rule out the effects of Ising SOC.

The system certainly breaks the mirror symmetries with mirror planes perpendicular to the samples. Therefore, the Ising SOC terms can arise. Even a few meV of band splitting due to Ising SOC can further enhance H_{c2} . The authors claimed that the Rashba SOC is dominant but this is only true for a small portion of the bands at the Fermi energy. The Rashba SOC can be very small in some parts of the bands. It is extremely difficult for DFT calculations to capture the few meV of band splitting due to Ising SOC. The presence of the Ising SOC terms can strongly enhance H_{c2} . Using the observed H_{c2} to estimate τ_s would produce an incorrect result and could result in the situation where τ_s is smaller than τ_{el} as shown in Table 1.

Authors:

Here, the reviewer raises two issues concerning the interpretation of the B_{c2} enhancement: reliability of the DFT calculations and the role of Zeeman (Ising) SOC. We refute them point-by-point in the following.

1) Reliability of the DFT calculations

We first address the reviewer's comment "*It is extremely difficult for DFT calculations to capture the few meV of band splitting due to Ising SOC*". Apart from the reliability of DFT calculations, the discussion of the band structure on this energy scale does not have much meaning here. This is because the Fermi surface of the present system should have energy broadening of 9–14 meV due to the electron elastic scattering even for flat samples, as discussed in the manuscript. In actual samples, the energy band splitting can exist only above this energy scale.

Our calculations are based on a widely adopted DFT package Quantum ESPRESSO. To check the reproducibility of our result, we have carried out the same calculation from scratch using another DFT package OpenMX. As shown below, the result by OpenMX is essentially the same as the one by Quantum ESPRESSO (Fig. A1(a)-(d)). We note that these two packages are based on completely different basis sets (OpenMX: pseudo-atomic orbitals, Quantum ESPRESSO: plane wave basis) and we used different pseudopotentials for the two calculations (OpenMX: norm-conserving pseudopotentials, Quantum ESPRESSO: projector augmented wave). Thus, the agreement between the two results indicates a high reliability of our calculations. We also note that calculated energy band structure reproduces our ARPES result remarkably well (see Fig. 2c). Furthermore, a recent spin-polarized ARPES experiment on the same system and an independent

DFT calculation using HiLAPW code (see T. Kobayashi et al., Phys. Rev. Lett. 125, 176401 (2020)) are also consistent with our calculations. Considering these facts, we conclude that our DFT result is highly reliable.

Figure A1: (a)(b) Fermi surface and Fermi velocity of $\sqrt{7}\times\sqrt{3}$ -In calculated by (a) Quantum ESPRESSO and (b) Open MX. (c)(d) Band splitting and spin polarization direction calculated by (c) Quantum ESPRESSO and (d) Open MX.

2) The role of Zeeman SOC

To investigate how strongly Zeeman SOC influences the spin directions, we calculated the distribution of spin polarization direction obtained from our DFT results (Fig. A2(a)). It clearly shows that the spins align in the in-plane directions for the most of energy regions. This means that Rashba SOC is dominant over Zeeman SOC mostly. The spins tend to tilt toward the out-of-plane direction below 30 meV, but the off-angle θ is about 45° at most. Namely, there is no region where Zeeman SOC is dominant. This result was also reproduced by the calculations using OpenMX (Fig. A2(b)).

Figure A2: Spin orientation angle θ relative to the in-plane direction as a function of energy splitting calculated using (a) Quantum ESPRESSO and (b) Open MX. $\theta = 0^\circ$ corresponds to the in-plane direction while $\theta = 90^\circ$ the out-of-plane direction (see the right illustration).

This non-dominant Zeeman SOC confined to small area of the Fermi surface cannot enhance B_{c2} strongly as the reviewer claims. This is because enhancement factor is determined by an average over the whole Fermi surface. Transition temperature in the presence of magnetic field \mathbf{B} , $T_c(\mathbf{B})$, is given by the following equations

$$\ln\left(\frac{T_c(\mathbf{B})}{T_{c0}}\right) = 2 \left\langle |\psi(\mathbf{k})|^2 f\left(\frac{\mathbf{g}(\mathbf{k}) \cdot \mathbf{B}}{\pi T_c |\mathbf{g}(\mathbf{k})|}\right) \right\rangle_{\mathbf{k}} \quad \dots (5)$$

$$f(x) = \text{Re} \sum_{n=1}^{\infty} \left(\frac{1}{2n-1+ix} - \frac{1}{2n-1} \right) \quad \dots (6)$$

where $T_{c0} \equiv T_c(\mathbf{B}=0)$, \mathbf{k} is the wavevector on the Fermi surface, $\psi(\mathbf{k})$ is the spin singlet gap function, $\mathbf{g}(\mathbf{k})$ is a vector determining the spin polarization at each \mathbf{k} , and $\langle \rangle_{\mathbf{k}}$ denotes taking an average over the Fermi surface (see M. Smidman et al, *Rep. Prog. Phys.* **80**, 036501 (2017)). B_{c2} is given by \mathbf{B} that satisfies $T_c(\mathbf{B})=0$. For example, when magnetic field is applied parallel to the Rashba-split Fermi surface, spins in some regions point nearly perpendicular to the field (see Fig. A3(a)). This is analogous to the Zeeman SOC case (see Fig. A3(b)), but the enhancement factor is limited to $\sqrt{2}$ because of the averaging over the whole Fermi surface. This clearly shows that, even if Zeeman SOC coexists in the present system and tilts some of the spins toward the out-of-plane direction, its effect is limited.

If the dynamics of spins is considered, the effect of the Zeeman SOC can be suppressed even more. Since the polarization axis of the spins strongly depends on momentum, electron scattering between different momenta effectively induces a spin flipping. Although some states on the Fermi surface have a spin Z component, it is not conserved through elastic scattering. Therefore, it cannot play a primary role in the enhancement of B_{c2} .

Based on all these facts, we conclude that strong enhancement of B_{c2} due to Zeeman SOC is extremely unlikely.

Figure A3: Schematic illustration of the Fermi surface splitting due to (a) Rashba SOC and (b) Zeeman SOC.

Actions:

New sentences, references, figures and notes have been added as follows.

- main text

(p.9)

“From the spin polarization direction calculated as a function of energy splitting, For more discussions, see Supplementary Note 4.”

(p.10)

“To check the reproducibility of our result, we carried out the same calculation from scratch using by Quantum ESPRESSO (See Supplementary Figures 5 and 6, as well as Supplementary Note 5).”

(p.15-16)

References 41, 47, 48

- supplementary file

Supplementary Figure 5

Supplementary Figure 6

Supplementary Note 4

Supplementary Note 5

Supplementary References 53-57

Reviewer #3 (Remarks to the Author):

I've read through the revised manuscript of NCOMMS-19-42030A, the answers to my questions, and also the replies to the comments from the other two referees. I felt that most of the points are addressed convincingly with sufficient improvements.

For the crucial point of spin-orbit scattering. I would suggest the authors to consider adding their discussion (rewrite it into a more concise form) below to the main text to highlight the essential role of Rashba SOC.

"It is true that spin-orbit scattering by Rashba-type SOC is not included in our model (or in the KLB model). Since the effect of the conventional spin-orbit scattering by the atomistic SOC on the H_c2 is widely known, we first analysed the data based on it without taking account of the Rashba-type SOC. In this way, we deduced successfully spin scattering time τ_s and compared it elastic electron scattering time τ_{el} . The result was surprising; we found $\tau_{el} = \tau_s$ within the experimental error for the flat samples and $\tau_{el} \sim 0.5\tau_s$ for the vicinal samples. If only the conventional spin-orbit scattering mechanism is considered, $\tau_{el}/\tau_s \sim (\alpha Z)^4 = 1/60$ for In, where α is the fine structure constant and Z is the atomic number ($Z = 49$). An experimental study reported even larger τ_{el}/τ_s of about $10-3$ for thin In films with a thickness of a few tens of nm [Physics Letters A 58, 131 (1976)]. Therefore, the spin-orbit scattering that occurs in the absence of Rashba SOC cannot account for the $\tau_{el}/\tau_s \sim 0.5-1$ obtained from our experiment. Nevertheless, if Rashba SOC is considered, this experimental result can be reasonably explained based on the concept of dynamic spin-momentum locking. This clearly shows why the Rashba splitting should be regarded as the primary origin of the large H_c2 ".

Authors:

We thank the reviewer for his/her high evaluation. According to the suggestion, we have incorporated the above argument into the manuscript to highlight the essential role of Rashba SOC.

Actions:

New sentences have been added as follows.

- main text

(p.8)

“Therefore, the spin-orbit scattering that occurs in the absence of the Rashba-type SOC cannot account for our result..... by orders of magnitude from the value expected for the conventional spin-orbit scattering.”

Other changes:

We have added T. Shishidou in the following sentence to acknowledge the discussion with him.

(p.16)

We thank Y. Higashi, S. Ichinokura and T. Shishidou for helpful discussions.

REVIEWERS' COMMENTS

Reviewer #3 (Remarks to the Author):

After reading the replies from the authors. I think the manuscript can be accepted for publication.

Reviewer #3 (Remarks to the Author):

Reviewer:

After reading the replies from the authors. I think the manuscript can be accepted for publication.

Authors:

We deeply thank the reviewer for reading our reply and evaluate our manuscript.